# MAA: Meticulous Adversarial Attack against Vision-Language Pre-trained Models

## Abstract

Current adversarial attacks for evaluating the robustness of vision-language pre-trained (VLP) models in multi-modal tasks suffer from limited transferability, where attacks crafted for a specific model often struggle to generalize effectively across different models, limiting their utility in assessing robustness more broadly. This is mainly attributed to the over-reliance on model-specific features and regions, particularly in the image modality. In this paper, we propose an elegant yet highly effective method termed Meticulous Adversarial Attack (MAA) to fully exploit model-independent characteristics and vulnerabilities of individual samples, achieving enhanced generalizability and reduced model dependence. MAA emphasizes fine-grained optimization of adversarial images by developing a novel resizing and sliding crop (RScrop) technique, incorporating a multi-granularity similarity disruption (MGSD) strategy. RScrop efficiently enriches the initial adversarial examples by generating more comprehensive, diverse, and detailed perspectives of the images, establishing a robust foundation for capturing representative and intrinsic visual characteristics. Building on this, MGSD seeks to maximize the embedding distance between adversarial examples and their original counterparts across different granularities and hierarchical levels within the architecture of VLP models, thereby amplifying the impact of the adversarial perturbations and enhancing the efficacy of attacks across every layer and component of the model. Extensive experiments across diverse VLP models, multiple benchmark datasets, and a variety of downstream tasks demonstrate that MAA significantly enhances the effectiveness and transferability of adversarial attacks. A large cohort of performance studies is conducted to generate insights into the effectiveness of various model configurations, guiding future advancements in this domain. The source code is provided in the supplementary material.

## 1 Introduction

Vision-language pre-trained (VLP) models have achieved remarkable success and serve as foundational models for a wide range of tasks, including information retrieval, image captioning, and visual question answering Radford et al. (2021); Li et al. (2022; 2021); Yang et al. (2022). These models are typically pre-trained on large-scale unlabeled datasets using self-supervised learning and subsequently fine-tuned for specific downstream tasks. Given their extensive applications, it is crucial to evaluate the robustness of VLP models to ensure their reliability in real-world scenarios, which are often characterized by uncertainties and potential threats. A representative method for assessing robustness is through adversarial attacks, where imperceptible perturbations are deliberately crafted to mislead models to wrongly associate images and texts, resulting in incorrect predictions.

Ensuring the transferability of adversarial attacks across different models is critical, as it is impractical to craft individual attacks for every different model in real-world scenarios, especially when attackers often lack access to target models. Existing methods usually enhance adversarial transferability by enlarging the feature distance between adversarial examples and their original counterparts across different modalities Zhang et al. (2022); Lu et al. (2023); He et al. (2023); Yin et al. (2023); Zhang et al. (2024). Some of them also use data augmentation techniques Lu et al. (2023); Zhang et al. (2024); He et al. (2023) to increase data diversity to further prevent overfitting to the target model during training (a.k.a. the source model). However, the performance of adversarial examples produced by these methods is less effective when applied to unknown target models, where adver-

Table 1: Comparison of different attack methods using image-only perturbations (im) and multi-modal perturbations (mul) in the image-text retrieval task on Flickr30K. Attack success rate (%) regarding the average of R@1 is used for evaluation. CLIP$_{\text{ViT-B/16}}$ is adopted as the source model. The grey background indicates the white-box attack results.

| Target Model | CLIP | | | | | | ALBEF | | TCL | |
|---|---|---|---|---|---|---|---|---|---|---|
| | ViT-B/16 | | ViT-L/14 | | RN101 | | | | | |
| Method | I2T | T2I | I2T | T2I | I2T | T2I | I2T | T2I | I2T | T2I |
| Co-Attack$_{\text{im}}$ | 90.55 | 91.72 | 7.48 | 16.4 | 8.94 | 12.42 | 2.82 | 5.78 | 5.16 | 7.98 |
| Co-Attack$_{\text{mul}}$ | 97.73 | 98.83 | 27.80 | 44.50 | 35.93 | 44.52 | 11.42 | 25.30 | 12.63 | 25.85 |
| SGA$_{\text{im}}$ | 98.04 | 99.00 | 15.58 | 23.29 | 15.33 | 21.20 | 5.11 | 10.41 | 7.06 | 12.12 |
| SGA$_{\text{mul}}$ | 99.53 | 99.73 | 32.14 | 47.83 | 44.01 | 51.19 | 14.86 | 29.56 | 16.26 | 30.66 |
| VLATTACK$_{\text{im}}$ | 99.88 | 99.97 | 8.34 | 16.24 | 13.41 | 17.87 | 2.92 | 7.74 | 6.11 | 10.07 |
| VLATTACK$_{\text{mul}}$ | 99.86 | 99.92 | 30.49 | 42.69 | 41.31 | 48.65 | 11.29 | 28.22 | 14.49 | 30.23 |

sarial perturbations for the image modality are extraordinarily less effective (an example is presented in Table 1). Their limited transferability stems from the failure to fully explore the characteristics and vulnerabilities of individual samples, as well as the over-reliance on model-specific patterns.

Transferability is particularly challenging in the context of VLP models due to two key factors: the unpredictable fine-tuning process and the complexity of involving multiple modalities. VLP models are usually fine-tuned for downstream tasks with task-specific optimizations, considering varying datasets, objectives and other factors. The complexity of multi-modalities introduces more intricate information. Models tend to extract any available information to make decisions, even those that do not accurately reflect the true semantic essence of data Zhang et al. (2021b); Ilyas et al. (2019); Hendrycks et al. (2021); Qin et al. (2022). The specialized factors of VLP models amplify this and make models more prone to relying on specific features and regions to associate images and texts. When these models are used as source models, attackers often generate adversarial examples that place greater emphasis on model-specific features and attended regions, resulting in overfitting to the source model and ultimately reducing transferability to other models.

In this paper, we focus on exploiting model-independent characteristics and vulnerabilities of images to guide the generation of adversarial examples, minimizing dependency on and susceptibility to the source models. A simple yet highly effective and transferable attack method is developed, termed the Meticulous Adversarial Attack (MAA). MAA refines adversarial examples primarily for the image modality to disrupt the understanding of image-text relationships across diverse models by augmenting low-level image details. Our approach is partially grounded in a well-acknowledged insight: *tailored perturbations to each individual image tend to be more potent than applying uniform perturbations across all images* Poursaeed et al. (2018); Naseer et al. (2019). With this in mind, MAA exploits representative and fine-grained characteristics and inherent vulnerabilities of original images to facilitate the generation of targeted perturbation. This approach substantially reduces over-reliance on the patterns/features that are generated specifically to the source model, thereby greatly enhancing the effectiveness and transferability of adversarial examples.

Specifically, we introduce a novel resizing and sliding crop (RScrop) technique, seamlessly integrated with a multi-granularity similarity disruption (MGSD) strategy. Essentially, similar to Yin et al. (2023); Ganeshan et al. (2019), the MGSD strategy enlarges the feature distances between adversarial examples and their original counterparts across various layers and components of the model. Low-level layers and components process local regions and detailed features, while high-level layers and components capture more abstract, semantic information. However, MGSD is restricted by the fixed-size input and local region processing in existing VLP models, which cannot effectively focus on more detailed aspects and their connections. For instance, in vision transformers, patch embedding is learnt using non-overlapping patches, where the boundary areas between adjacent patches are often ignored, resulting in the loss of crucial contextual information. This limitation prevents the model from capturing fine-grained local dependencies or recognizing patterns that span across patch borders. Consequently, this lack of continuity can hinder the model's ability to represent subtle textures, edges, and complex spatial relationships present in the image. Though the kernels applied in CNNs can slide across an image with overlapping areas, their fixed size and localized receptive fields inherently limit their ability to capture more local information. To capture a fine-grained as well as comprehensive view of images to distill their representative

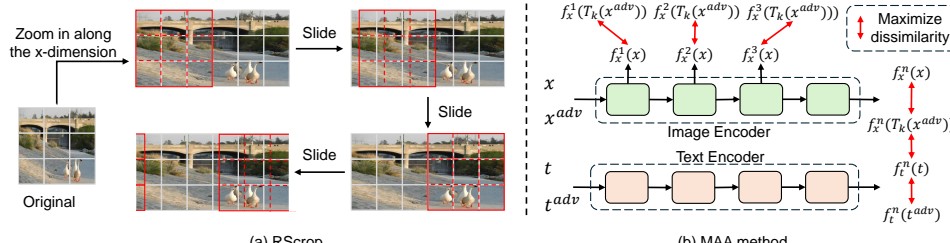

Figure 1: An illustration of the proposed (a) RScrop augmentation and (b) MAA method. For RScrop, we use the ViT-based model as an example, where images are processed patch-wise. RScrop zooms in on adversarial examples and applies them in a sliding manner along each dimension (with the x-dimension as an example) to capture more fine-grained local regions and their interconnections. The crop window shifts by a small step size and moves to adjacent non-overlapping areas relative to its previous position, repeating this process until the entire image is covered. MAA enlarges the feature distance between adversarial images and the original images across various layers and components of the model, while also maximizing the cross-modal gap.

characteristics for performing an effective MGSD, RScrop is proposed, supported by scale- and translation-invariant properties of DNNs Lin et al. (2019); Dong et al. (2019). It employs resizing and cropping operations, to scale up adversarial examples and feed them in a sliding manner into the model to enable exploration of more fine-grained features and local regions. We also maximize the embedding distance between image and text pairs to widen the modality gap for generating both adversarial images and texts.

The benefits of this approach are manifold. First, augmenting through scaling and sliding acts as a magnifier, enabling the model to attend to intricate local details and previously overlooked boundary regions of adjacent patches in individual images. This meticulous focus on fine-grained elements enhances spatial coherence and captures subtle variations, ultimately improving the model's sensitivity to nuanced patterns and contextual dependencies within the image itself, independent of model- and task-specific objectives. Generating adversarial examples based on these augmented data can successfully alleviate the reliance on model-specific patterns, therefore relieving the overfitting issue. Second, intermediate features especially low-level features are generally more generalized and shared across various models. As a result, MAA promotes more sample-dependent and model-generic adversarial examples, improving transferability. Third, extensive experiments demonstrate that MAA achieves notable improvements over existing state-of-the-art techniques in terms of adversarial transferability. Last but not least, sophisticated parameter studies are undertaken to provide a comprehensive and in-depth analysis of model performance, shedding light on critical design choices and fostering the development of more refined models in future work.

## 2 METHODOLOGY

### 2.1 PRELIMINARIES

Let $F_s$ represent an available source VLP model consisting of an image encoder $f_{\text{img}}$ and a text encoder $f_{\text{txt}}$, which learn feature representations for images and texts, respectively. Given an image-text pair $(\boldsymbol{x}, \boldsymbol{t})$, the objective is to generate adversarial examples $\boldsymbol{x}^{\text{adv}}$ and $\boldsymbol{t}^{\text{adv}}$ that can mislead the predictions of an unknown target model $F_t$. To ensure the perturbations remain imperceptible, for image perturbations, we use $l_\infty$-norm constraint: $||\boldsymbol{t}^{\text{adv}} - \boldsymbol{t}||_\infty \leq \epsilon_{\text{img}}$. For text perturbations, we restrict the number of words that can be modified in the sentence, denoted as $\epsilon_{\text{txt}}$.

### 2.2 METICULOUS ADVERSARIAL ATTACK

MAA employs a straightforward RScrop technique in conjunction with a multi-granularity similarity disruption (MGSD) strategy. An illustration of the framework can be found in Figure 1.

The MGSD strategy enhances adversarial images by enlarging the feature distance from the original images across various layers and components of the model. Specifically, low-level layers and com-

ponents process local regions and extract detailed features, e.g., patch processing of ViTs, which enable us to enhance the local utility of adversarial examples. High-level ones capture abstract, semantic information. By targeting these diverse granularities and hierarchical levels, we can effectively uncover the characteristics and vulnerabilities of samples and also make the generation of adversarial examples more rely on samples and less reliant on information specialized for source models to associate images and tests. Moreover, low-level layers and components are generally less model-specific. These contribute to reducing overfitting and enhancing the transferability. However, common backbone networks used in VLP models, such as ViTs and CNNs, are constrained by fixed-size processing of inputs and limited local regions. Relying solely on these networks to generate adversarial examples fails to fully explore the representative and intrinsic characteristics and vulnerability of samples. Specifically, these networks process fixed-size inputs, which can hinder the understanding of features of local regions. While targeting low-level layers that focus on local regions can alleviate this, it does not completely solve the problem. For instance, ViTs process images in non-overlapping patches, which overlook important local relationships between neighboring patches. Similarly, CNNs utilize fixed-size convolutional filters, also imposing restrictions on feature exploration. Furthermore, they cannot extract local information from high-level layers and components, limiting the fine-grained optimization of adversarial samples. As different models are prone to extract some distinct features and focus on different regions to associate images and texts, the fixed-size constraints would make generated adversarial examples rely on model-specific features and regions, hindering the transferability across different models.

To overcome fixed-size constraints and explore more comprehensive, diverse, and detailed information, we propose RScrop, which involves two operations: scaling and cropping. As illustrated in Figure 1(a). First, original adversarial images are scaled up along each dimension, enabling a focus on fine-grained local regions. Second, we systematically crop local regions by sliding a crop window across the image, starting from an initial point in each dimension. The window shifts by a predefined amount along each dimension to ensure comprehensive coverage of the image with more regions and their connections considered. Specifically, after the initial crop, the window is shifted by a step size randomly selected to be smaller than the dimension size of the patch or the convolutional filter of the first layer, ensuring that more local regions and their connections are considered. We then move to adjacent non-overlapping areas from the previous crop (excluding the small-step cropped regions), applying the same operation until the entire image is covered. (R4W2) The shift step length for $i$-th step relative to the initial point in different dimensions can be formulated by $L^i_{x/y} = (i/2) * l_{x/y} + (i\%2) * \alpha_{x/y}(i), \alpha^i_{x/y} = UniformDiscrete(\beta_1, \beta_2)$, where $x$ and $y$ denote $x$- and $y$- dimension, and $\beta_1$ and $\beta_2$ are smaller than the size of dimension size of the patch or the convolutional filter, i.e., $l_{x/y}$. Supported by scale- and translation-invariant properties of DNNs Lin et al. (2019); Dong et al. (2019), this method ensures thorough processing of local areas and their relationships, providing complete coverage of the image across various layers and complements.

(R2W3,R4W2) Formally, after Rscrop, we can obtain a set of transformed adversarial images $\mathcal{R}(\boldsymbol{x}^{\text{adv}}) = \{\boldsymbol{x}^{\text{adv}}_1, \boldsymbol{x}^{\text{adv}}_2, ..., \boldsymbol{x}^{\text{adv}}_k\}$. We maximize the feature distance between all these adversarial images and original images at various layers and components as follows :

$$(\text{R2W3,R4W2}) \min_{\boldsymbol{x}^{\text{adv}}} \mathcal{L}_1 = \sum_{i=1}^{N} \sum_{x' \in \mathcal{R}(\boldsymbol{x}^{\text{adv}})} (\cos(f^i_{\text{img}}(\boldsymbol{x}'), f^i_{\text{img}}(\boldsymbol{x}))), \tag{1}$$

where $N$ is the number of layers and components, and $f^i_{\text{img}}$ is the representations output from the $i$-th layer or component. (R4W2) For ViT-based VLP models, features would include those output from self-attention modules and final output layer, while for ResNet-based VLP models, we extract features from residual blocks and final output layer. By focusing on features across different layers and different scales of images, we are able to explore fine-grained vulnerabilities of samples.

(R4W2) The RScrop also creates diverse image-text pairs, which help better explore cross-model interactions for transferrable attacks. We enlarge the feature distance between adversarial images and their original paired texts to comprehensively disturb image-text connections:

$$\min_{\boldsymbol{x}^{\text{adv}}} \mathcal{L}_2 = \sum_{i=1}^{N} \sum_{x' \in \mathcal{R}(\boldsymbol{x}^{\text{adv}})} (\cos(f_{\text{img}}(\hat{\boldsymbol{x}}'_k), f_{\text{txt}}(\boldsymbol{t}))). \tag{2}$$

The overall objective for learning adversarial images is as follows:

$$\min_{\boldsymbol{x}^{\mathrm{adv}}} \mathcal{L}_{\mathrm{img}} = \mathcal{L}_1 + \mathcal{L}_2. \tag{3}$$

For attacking the text modality, we use the commonly used BERT-Attack method Li et al. (2020) to generate adversarial texts by maximizing the feature distance from their original image-text pairs:

$$\min_{\boldsymbol{t}^{\mathrm{adv}}} \mathcal{L}_{\mathrm{txt}} = \cos(f_{\mathrm{txt}}(\boldsymbol{t}^{\mathrm{adv}}), f_{\mathrm{txt}}(\boldsymbol{t})) + \cos(f_{\mathrm{txt}}(\boldsymbol{t}^{\mathrm{adv}}), f_{\mathrm{img}}(\boldsymbol{x})). \tag{4}$$

(R4W2,R5W2) BERT-Attack first identifies the most important word in each sentence by replacing each word in the sentence with [MASK] one at a time and ranking the feature distance between each modified sentence with the original sentence and the paired image. The most important word would be replaced by a semantic-consistent word to ensure visually plausible attacks. For the attack effectiveness, BERT is used to generate a set of candidates and the one that can fulfil Eq. 4 would be selected to replace the original word to realize an attack.

## 3 EXPERIMENTS

### 3.1 SETTINGS

**Datasets and Tasks.** Three datasets are used: Flickr30K Plummer et al. (2015), MSCOCO Lin et al. (2014), and RefCOCO+ Yu et al. (2016). The splits for training and testing follow recent works Zhang et al. (2022); Lu et al. (2023); Yin et al. (2023); Zhang et al. (2024). We focus on three tasks: image-text retrieval, image captioning, and visual grounding. For image-text retrieval, Flickr30K Plummer et al. (2015) and MSCOCO Lin et al. (2014) are used. For the image captioning task and the visual grounding task, MSCOCO and RefCOCO+ datasets are used, respectively.

**Models.** We test MAA and all compared methods on four kinds of widely-used VLP models: CLIP Radford et al. (2021), ALBEF Li et al. (2021), TCL Yang et al. (2022), and BLIP Li et al. (2022). For CLIP, different image encoders are utilized, including vision transformers (ViT-B/16 and ViT-L/14 Dosovitskiy et al. (2020)) and CNNs (ResNet50 and ResNet101 He et al. (2016)). The text encoder is a 6-layer transformer. For ALBEF and BLIP, we choose the variant that consists of a ViT-B/16 as the image encoder and a 6-layer transformer as the text encoder for the attack.

**Implementation details.** For fundamental experiments, the perturbation magnitude is set to $\epsilon_x = 4/255$ for images and $\epsilon_t = 1$ for text. MAA is also evaluated across various perturbation magnitudes. The optimization problem for image perturbations is addressed using Projected Gradient Descent (PGD) Madry et al. (2018) with $T = 60$ iterations and a step size of $\epsilon_x/T \times 2.25$. The batch size is configured to 4. Every 10 iterations, scaling ratios would be changed by randomly selecting from the set $\{1.25, 1.5, 1.75, 2\}$ for each dimension to optimize the adversarial images.

**Baselines.** Several state-of-the-art approaches are selected for comparison, including Co-Attack Zhang et al. (2022), SGA Lu et al. (2023), VLATTACK Yin et al. (2023), ETU Zhang et al. (2024), and (R5Q1)VLPTransferAttack Gao et al. (2024). ETU only generates universal adversarial images, while others produce adversarial images and texts. We also include baseline methods such as PGD and BERT-Attack Li et al. (2020).

**Evaluation metric.** In image-text retrieval, the attack success rate (ASR) is used to evaluate all methods, calculated as the percentage of adversarial examples that successfully deceive the model. For other tasks, we compare the performance of target models before and after attacks.

### 3.2 TASK ANALYSIS

#### 3.2.1 RESULTS ON THE IMAGE-TEXT RETRIEVAL

Image-text retrieval involves ranking the similarity between queries and data to return the most relevant results. Aligning with aforementioned experiment settings, we assess the transferability of all methods in this task across various datasets (i.e., Flickr30K and MSCOCO) and VLP models (i.e., CLIP$_{\mathrm{ViT\text{-}B/16}}$, CLIP$_{\mathrm{ViT\text{-}L/14}}$, CLIP$_{\mathrm{ResNet50}}$, CLIP$_{\mathrm{ResNet101}}$, ALBEF, and TCL) for both image-to-text (I2T) and text-to-image (T2I) retrieval. Adversarial images are resized to meet the input requirements of each model. For instance, adversarial images generated on CLIP models are resized from

Table 2: The attack success rate (%) on the image-text retrieval task. CLIP$_{ViT-B/16}$ is adopted as the source model for training, while the target models include CLIP$_{ViT-B/16}$, CLIP$_{ViT-L/14}$, CLIP$_{ResNet50}$, CLIP$_{ResNet101}$, ALBEF, and TCL. The grey background indicates the white-box attack results. **Bold** indicates the best results.

| Dataset | | Flickr30K | | | | | | MSCOCO | | | | | |
| --- | --- | --- | --- | --- | --- | --- | --- | --- | --- | --- | --- | --- | --- |
| | | Image-to-Text | | | Text-to-Image | | | Image-to-Text | | | Text-to-Image | | |
| Target Model | Method | R@1 | R@5 | R@10 | R@1 | R@5 | R@10 | R@1 | R@5 | R@10 | R@1 | R@5 | R@10 |
| CLIP$_{ViT-B/16}$ | (R4Q2)PGD | 90.55±0.89 | 78.09±0.52 | 78.42±0.34 | 98.68±0.82 | 95.83±0.68 | 94.23±0.56 | 97.44±0.11 | 94.53±0.24 | 91.06±0.23 | 96.53±0.26 | 91.90±0.33 | 88.57±0.49 |
| | (R4Q2)BERT-Attack | 28.34±0.00 | 11.63±0.00 | 6.71±0.00 | 39.05±0.00 | 24.06±0.00 | 17.40±0.00 | 55.25±0.00 | 37.26±0.00 | 28.93±0.00 | 57.86±0.00 | 45.05±0.00 | 38.73±0.00 |
| | (R4Q2)Co-Attack | 97.73±0.18 | 94.29±0.02 | 92.08±0.40 | 98.83±0.05 | 96.17±0.05 | 94.38±0.08 | 99.50±0.12 | 98.75±0.10 | 97.86±0.35 | 99.60±0.06 | 98.89±0.05 | 98.25±0.12 |
| | (R4Q2)SGA | 99.53±0.05 | 97.96±0.03 | 96.48±0.08 | 99.73±0.03 | 98.88±0.03 | 97.72±0.12 | 91.91±0.02 | 82.96±0.21 | 78.69±0.09 | 91.18±0.12 | 81.57±0.32 | 79.93±0.08 |
| | (R4Q2)ETU | 97.40±0.09 | 94.16±0.54 | 78.84±0.32 | 90.24±0.22 | 84.31±0.26 | 69.32±0.28 | 91.56±0.19 | 80.45±0.21 | 78.87±0.37 | 90.06±0.24 | 83.63±0.09 | 86.17±0.43 |
| | (R4Q2)VLATTACK | 99.86±0.14 | **99.99±0.01** | 99.50±0.23 | 99.92±0.05 | 99.54±0.24 | **99.56±0.06** | 99.93±0.07 | 99.79±0.11 | 99.75±0.06 | **99.98±0.02** | **99.96±0.04** | **99.89±0.06** |
| | (R5Q1)VLPTransferAttack | **99.98±0.02** | 99.53±0.03 | 99.42±0.07 | **99.99±0.01** | **99.76±0.02** | 99.53±0.01 | **99.93±0.03** | **99.90±0.01** | 99.77±0.02 | 99.95±0.03 | 99.88±0.02 | 99.80±0.02 |
| | (R4Q2)MAA | 99.75±0.02 | 99.05±0.02 | 98.65±0.07 | 99.66±0.05 | 99.25±0.06 | 98.85±0.04 | **99.93±0.05** | 99.81±0.07 | **99.92±0.02** | 99.90±0.05 | 99.70±0.26 | 99.79±0.11 |
| CLIP$_{ViT-L/14}$ | (R4Q2)PGD | 7.56±0.42 | 2.02±0.38 | 0.59±0.18 | 16.71±0.56 | 6.27±0.53 | 3.66±0.29 | 20.25±0.52 | 10.19±0.28 | 7.04±0.09 | 29.10±0.31 | 16.13±0.24 | 12.49±0.18 |
| | (R4Q2)BERT-Attack | 24.79±0.00 | 9.76±0.00 | 5.59±0.00 | 39.05±0.00 | 23.01±0.00 | 17.07±0.00 | 51.05±0.00 | 32.89±0.00 | 25.82±0.00 | 59.63±0.00 | 44.27±0.00 | 37.56±0.00 |
| | (R4Q2)Co-Attack | 27.80±0.18 | 10.44±0.06 | 5.29±0.10 | 44.50±0.64 | 27.31±0.35 | 19.93±0.04 | 53.58±0.32 | 36.24±0.26 | 28.29±0.31 | 64.28±0.17 | 48.89±0.20 | 41.99±0.15 |
| | (R4Q2)SGA | 32.14±0.28 | 15.52±0.12 | 9.63±0.22 | 47.83±0.06 | 30.21±0.13 | 23.17±0.22 | 57.28±0.11 | 41.33±0.46 | 33.28±0.17 | 65.98±0.21 | 50.89±0.22 | 44.03±0.41 |
| | (R4Q2)ETU | 8.22±0.47 | 1.12±0.08 | 0.20±0.01 | 16.24±0.32 | 5.45±0.26 | 3.46±0.32 | 20.22±0.09 | 10.46±0.15 | 7.92±0.06 | 25.53±0.18 | 14.58±0.16 | 10.44±0.20 |
| | (R4Q2)VLATTACK | 30.49±0.06 | 12.54±0.38 | 6.04±0.09 | 42.69±0.28 | 26.64±0.19 | 19.32±0.25 | 56.22±0.23 | 37.39±0.19 | 29.91±0.43 | 61.55±0.53 | 45.28±0.37 | 37.12±0.32 |
| | (R4Q2,R5Q1)VLPTransferAttack | 42.54±0.52 | 23.99±0.23 | 14.55±0.41 | 53.82±0.05 | 36.24±0.19 | 28.39±0.27 | 64.82±0.28 | 48.98±0.19 | 40.54±0.06 | 71.85±0.24 | 58.66±0.30 | 51.77±0.34 |
| | (R4Q2)MAA | **54.52±0.31** | **33.36±0.12** | **25.76±0.16** | **62.80±0.50** | **46.32±0.27** | **38.11±0.21** | **80.72±0.13** | **67.83±0.14** | **61.75±0.73** | **83.42±0.49** | **72.67±0.36** | **67.03±0.21** |
| CLIP$_{ResNet50}$ | (R4Q2)PGD | 14.89±0.46 | 4.884±0.51 | 2.02±0.27 | 24.23±0.39 | 9.05±0.24 | 5.56±0.16 | 29.03±0.33 | 15.39±0.20 | 10.88±0.15 | 38.12±0.56 | 22.64±0.36 | 16.99±0.23 |
| | (R4Q2)BERT-Attack | 35.25±0.00 | 14.69±0.00 | 8.03±0.00 | 44.73±0.00 | 27.7±0.00 | 20.28±0.00 | 59.05±0.00 | 42.18±0.00 | 32.88±0.00 | 67.43±0.00 | 52.64±0.00 | 45.84±0.00 |
| | (R4Q2)Co-Attack | 39.35±0.44 | 17.98±0.64 | 10.03±0.19 | 50.54±0.61 | 31.64±0.78 | 25.21±0.23 | 64.00±0.33 | 45.92±0.31 | 37.29±0.24 | 71.95±0.20 | 56.88±0.24 | 49.78±0.17 |
| | (R4Q2)SGA | 43.00±0.71 | 22.88±0.23 | 15.65±0.34 | 56.78±0.12 | 36.47±0.24 | 28.89±0.31 | 68.78±0.46 | 53.01±0.16 | 43.76±0.28 | 75.46±0.19 | 62.00±0.22 | 54.58±0.16 |
| | (R4Q2)ETU | 12.45±0.26 | 3.46±0.05 | 2.35±0.43 | 22.70±0.15 | 7.37±0.28 | 5.06±0.02 | 27.67±0.37 | 15.11±0.16 | 10.90±0.12 | 37.12±0.16 | 21.79±0.17 | 16.10±0.09 |
| | (R4Q2)VLATTACK | 44.32±0.22 | 20.80±0.08 | 14.26±0.31 | 54.31±0.31 | 35.07±0.08 | 27.78±0.26 | 70.28±0.15 | 52.14±0.19 | 43.52±0.33 | 75.54±0.47 | 60.99±0.39 | 53.35±0.62 |
| | (R5Q1)VLPTransferAttack | 52.65±0.41 | 31.48±0.24 | 22.47±0.22 | 63.34±0.46 | 43.52±0.25 | 35.53±0.32 | 76.35±0.37 | 61.39±0.22 | 53.23±0.25 | 80.82±0.36 | 68.29±0.42 | 70.10±0.23 |
| | (R4Q2)MAA | **77.02±0.25** | **63.62±0.64** | **52.85±0.45** | **80.26±0.43** | **65.42±0.56** | **58.12±0.24** | **92.28±0.60** | **85.71±0.52** | **80.46±0.82** | **92.86±0.52** | **86.32±0.37** | **82.03±0.29** |
| CLIP$_{ResNet101}$ | (R4Q2)PGD | 8.84±0.67 | 2.03±0.34 | 1.30±0.27 | 12.82±0.47 | 4.28±0.28 | 2.68±0.10 | 14.89±0.09 | 7.92±0.13 | 5.85±0.06 | 21.07±0.06 | 11.58±0.32 | 8.91±0.12 |
| | (R4Q2)BERT-Attack | 30.27±0.00 | 11.63±0.00 | 5.77±0.00 | 37.39±0.00 | 24.92±0.00 | 18.59±0.00 | 52.39±0.00 | 35.55±0.00 | 28.76±0.00 | 58.64±0.00 | 46.29±0.00 | 39.39±0.00 |
| | (R4Q2)Co-Attack | 35.93±0.48 | 13.80±0.11 | 8.75±0.12 | 44.52±0.45 | 29.13±0.23 | 22.88±0.19 | 58.87±0.74 | 40.41±0.54 | 32.89±0.37 | 65.54±0.13 | 52.18±0.16 | 45.17±0.14 |
| | (R4Q2)SGA | 44.01±0.29 | 21.87±0.31 | 14.31±0.44 | 51.19±0.38 | 33.25±0.51 | 26.65±0.33 | 61.83±0.29 | 47.87±0.35 | 40.35±0.38 | 69.01±0.44 | 56.29±0.46 | 49.65±0.27 |
| | (R4Q2)ETU | 7.89±0.62 | 1.97±0.32 | 1.15±0.06 | 12.45±0.14 | 4.75±0.07 | 2.88±0.15 | 18.24±0.62 | 12.62±0.24 | 8.70±0.27 | 23.01±0.44 | 10.15±0.20 | 9.08±0.09 |
| | (R4Q2)VLATTACK | 41.31±0.28 | 18.42±0.17 | 11.03±0.25 | 48.65±0.10 | 31.47±0.19 | 24.65±0.15 | 62.88±0.09 | 46.81±0.05 | 38.80±0.20 | 70.34±0.32 | 56.43±0.31 | 49.45±0.25 |
| | (R5Q1)VLPTransferAttack | 51.64±0.56 | 28.45±0.19 | 19.74±0.32 | 59.42±0.22 | 40.83±0.35 | 33.21±0.44 | 70.86±0.42 | 57.47±0.29 | 48.63±0.24 | 76.49±0.37 | 64.39±0.17 | 57.58±0.25 |
| | (R4Q2)MAA | **72.23±0.82** | **55.34±0.37** | **44.89±0.41** | **74.87±0.23** | **60.65±0.42** | **53.42±0.39** | **89.56±0.49** | **81.35±0.27** | **76.86±0.38** | **90.07±0.16** | **83.12±0.02** | **78.66±0.06** |
| ALBEF | (R4Q2)PGD | 2.87±0.09 | 0.23±0.05 | 0.30±0.03 | 5.85±0.21 | 1.83±0.11 | 1.10±0.08 | 8.42±0.49 | 3.87±0.53 | 1.87±0.08 | 13.11±0.36 | 5.49±0.12 | 3.50±0.12 |
| | (R4Q2)BERT-Attack | 9.45±0.00 | 1.31±0.00 | 0.45±0.00 | 22.82±0.00 | 11.22±0.00 | 8.15±0.00 | 28.32±0.00 | 12.67±0.00 | 8.25±0.00 | 41.16±0.00 | 26.17±0.00 | 20.77±0.00 |
| | (R4Q2)Co-Attack | 11.42±0.22 | 1.92±0.18 | 0.53±0.08 | 25.30±0.16 | 12.71±0.13 | 9.33±0.13 | 31.01±0.44 | 14.29±0.60 | 9.37±0.43 | 43.34±0.73 | 27.71±0.61 | 22.17±0.51 |
| | (R4Q2)SGA | 14.86±0.05 | 3.31±0.00 | 1.40±0.00 | 29.56±0.02 | 14.32±0.24 | 10.68±0.32 | 36.62±0.20 | 18.93±0.08 | 13.13±0.24 | 46.78±0.19 | 30.74±0.19 | 24.05±0.31 |
| | (R4Q2)ETU | 1.72±0.05 | 0.30±0.00 | 0.20±0.00 | 5.89±0.46 | 1.26±0.09 | 0.88±0.03 | 12.83±0.47 | 6.86±0.26 | 3.34±0.38 | 13.16±0.67 | 7.23±0.37 | 4.34±0.16 |
| | (R4Q2)VLATTACK | 11.29±0.15 | 2.52±0.10 | 1.00±0.00 | 28.22±0.42 | 13.78±0.15 | 10.23±0.11 | 34.60±0.17 | 17.21±0.29 | 11.25±0.24 | 46.73±0.33 | 31.43±0.42 | 25.34±0.41 |
| | (R5Q1)VLPTransferAttack | 30.28±0.29 | 12.96±0.16 | 6.14±0.35 | 42.88±0.08 | 25.45±0.12 | 19.17±0.09 | 52.23±0.23 | 30.39±0.07 | 24.74±0.12 | 59.27±0.24 | 44.09±0.26 | 34.21±0.19 |
| | (R4Q2)MAA | **32.45±0.67** | **14.68±0.43** | **11.20±0.21** | **43.62±0.55** | **25.63±0.24** | **19.83±0.20** | **53.82±0.52** | **33.38±0.23** | **26.01±0.54** | **60.78±0.74** | **44.35±0.29** | **36.44±0.25** |
| TCL | (R4Q2)PGD | 5.28±0.32 | 0.32±0.05 | 0.10±0.02 | 8.10±0.22 | 2.31±0.19 | 1.39±0.12 | 10.51±0.27 | 4.19±0.22 | 2.40±0.07 | 14.62±0.31 | 6.45±0.22 | 4.09±0.12 |
| | (R4Q2)BERT-Attack | 9.59±0.00 | 2.01±0.00 | 0.60±0.00 | 24.05±0.00 | 11.89±0.00 | 8.20±0.00 | 29.15±0.00 | 13.45±0.00 | 9.39±0.00 | 41.01±0.00 | 25.96±0.00 | 19.93±0.00 |
| | (R4Q2)Co-Attack | 12.63±0.44 | 3.09±0.12 | 0.84±0.19 | 25.85±0.56 | 13.89±0.58 | 9.29±0.33 | 32.35±0.32 | 14.78±0.51 | 10.50±0.23 | 43.72±0.19 | 27.77±0.33 | 21.86±0.12 |
| | (R4Q2)SGA | 16.26±0.21 | 3.59±0.16 | 1.47±0.05 | 30.66±0.27 | 16.05±0.19 | 11.20±0.15 | 37.14±0.28 | 18.98±0.32 | 12.87±0.08 | 47.05±0.16 | 31.18±0.11 | 24.79±0.21 |
| | (R4Q2)ETU | 11.78±0.68 | 6.12±0.36 | 2.06±0.11 | 17.54±0.43 | 8.65±0.44 | 6.08±0.23 | 17.36±0.28 | 5.55±0.34 | 3.92±0.14 | 15.72±0.26 | 7.26±0.33 | 5.50±0.21 |
| | (R4Q2)VLATTACK | 14.49±0.25 | 3.71±0.19 | 1.47±0.12 | 30.23±0.22 | 16.34±0.43 | 11.22±0.32 | 35.94±0.18 | 18.85±0.09 | 12.67±0.34 | 48.39±0.28 | 32.34±0.15 | 25.87±0.18 |
| | (R5Q1)VLPTransferAttack | 30.66±0.09 | 12.79±0.16 | 6.10±0.12 | 42.79±0.45 | 26.52±0.33 | 19.20±0.18 | 56.24±0.28 | 37.39±0.22 | 29.58±0.34 | 61.58±0.27 | 45.10±0.03 | 36.96±0.11 |
| | (R4Q2)MAA | **41.42±0.48** | **22.42±0.11** | **17.03±0.15** | **50.29±0.41** | **31.24±0.25** | **24.82±0.33** | **60.95±0.39** | **39.96±0.22** | **31.65±0.36** | **62.37±0.27** | **45.17±0.24** | **37.46±0.28** |

Table 3: The attack success rate (%) of R@1 in image-text retrieval on Flickr30K. Grey background highlights white-box attack results, and **bold** indicates the best performance.

| Target Model | | CLIP | | | | | | | | ALBEF | | TCL | |
| --- | --- | --- | --- | --- | --- | --- | --- | --- | --- | --- | --- | --- | --- |
| | | ViT-B/16 | | ViT-L/14 | | RN50 | | RN101 | | | | | |
| Source Model | Method | I2T | T2I | I2T | T2I | I2T | T2I | I2T | T2I | I2T | T2I | I2T | T2I |
| CLIP$_{ResNet101}$ | (R4Q2)PGD | 2.22±0.17 | 6.73±0.09 | 8.42±0.12 | 15.91±0.12 | 15.28±0.22 | 23.76±0.19 | 93.27±0.06 | 96.62±0.18 | 1.78±0.17 | 4.45±0.11 | 3.82±0.15 | 6.95±0.25 |
| | (R4Q2)BERT-Attack | 27.12±0.00 | 37.44±0.00 | 25.28±0.00 | 39.85±0.00 | 35.52±0.00 | 46.91±0.00 | 30.52±0.09 | 39.97±0.00 | 9.49±0.00 | 23.25±0.00 | 11.59±0.00 | 24.36±0.00 |
| | (R4Q2)Co-Attack | 28.67±0.27 | 40.42±0.33 | 25.58±0.16 | 41.68±0.19 | 38.62±0.46 | 52.89±0.18 | 98.18±0.13 | 98.57±0.06 | 9.39±0.12 | 23.83±0.24 | 11.82±0.16 | 25.18±0.18 |
| | (R4Q2)SGA | 32.57±0.11 | 44.37±0.24 | 28.32±0.09 | 43.19±0.06 | 49.27±0.22 | 61.28±0.27 | 97.85±0.11 | 97.86±0.09 | 11.82±0.23 | 26.51±0.17 | 14.69±0.12 | 28.79±0.27 |
| | (R4Q2)ETU | 5.28±0.09 | 7.43±0.17 | 7.63±0.26 | 10.15±0.08 | 13.64±0.23 | 16.40±0.17 | 88.87±0.22 | 89.52±0.36 | 4.31±0.21 | 7.36±0.08 | 5.71±0.22 | 8.98±0.10 |
| | (R4Q2)VLATTACK | 33.00±0.31 | 43.27±0.09 | 27.80±0.17 | 40.72±0.15 | 48.74±0.08 | 59.72±0.18 | 94.25±0.24 | 96.82±0.07 | 3.92±0.09 | 9.38±0.13 | 8.68±0.26 | 13.91±0.31 |
| | (R4Q2)VLPTransferAttack | 35.28±0.45 | 46.51±0.23 | 28.41±0.15 | 43.65±0.29 | 60.54±0.32 | 69.71±0.17 | 94.13±0.08 | 96.78±0.14 | 12.39±0.34 | 27.98±0.14 | 16.65±0.19 | 30.07±0.09 |
| | (R4Q2)MAA | **36.03±0.81** | **47.56±0.52** | **30.06±0.86** | **44.80±0.66** | **70.83±0.45** | **78.93±0.48** | **98.98±0.82** | **98.82±1.01** | **14.08±0.52** | **29.15±0.15** | **17.01±0.26** | **31.94±0.42** |
| ALBEF | (R4Q2)PGD | 10.45±0.54 | 14.92±0.24 | 16.52±0.14 | 30.83±0.32 | 15.50±0.11 | 23.79±0.37 | 22.27±0.23 | 38.24±0.16 | 87.12±0.08 | 90.07±0.10 | 25.04±0.23 | 30.12±0.17 |
| | (R4Q2)BERT-Attack | 25.03±0.00 | 35.66±0.00 | 28.88±0.00 | 38.04±0.00 | 37.12±0.00 | 56.90±0.00 | 31.42±0.00 | 45.35±0.00 | 32.53±0.00 | 50.61±0.00 | 8.22±0.00 | 19.57±0.00 |
| | (R4Q2)Co-Attack | 27.88±0.25 | 38.52±0.09 | 24.23±0.61 | 38.51±0.23 | 33.26±0.19 | 46.42±0.10 | 33.58±0.42 | 46.39±0.16 | 98.61±0.08 | 98.66±0.05 | 28.25±0.15 | 42.99±0.51 |
| | (R4Q2)SGA | 38.76±0.27 | 47.45±0.25 | 31.20±0.07 | 45.65±0.32 | 46.57±0.28 | 58.10±0.12 | 42.23±0.28 | 51.40±0.41 | 98.69±0.05 | 98.54±0.07 | 64.45±0.71 | 69.11±0.22 |
| | (R4Q2)ETU | 12.32±0.46 | 17.73±0.22 | 15.67±0.51 | 18.42±0.12 | 16.18±0.25 | 25.69±0.33 | 22.90±0.07 | 28.57±0.23 | 87.06±0.28 | 89.33±0.09 | 28.23±0.46 | 32.38±0.27 |
| | (R4Q2)VLATTACK | 36.42±0.33 | 45.86±0.18 | 29.13±0.25 | 44.62±0.11 | 40.02±0.23 | 54.12±0.19 | 35.60±0.12 | 47.09±0.27 | 94.12±0.08 | 96.80±0.06 | 41.81±0.54 | 52.69±0.29 |
| | (R4Q2)VLPTransferAttack | 38.93±0.18 | 48.90±0.12 | 30.50±0.32 | 45.12±0.11 | 48.11±0.09 | 58.64±0.32 | 42.53±0.27 | 52.37±0.76 | 99.74±0.09 | **99.72±0.12** | 74.24±0.36 | 74.48±0.17 |
| | (R4Q2)MAA | **39.70±0.58** | **50.09±0.35** | **32.52±0.14** | **47.50±0.24** | **51.02±0.08** | **61.07±0.11** | **44.34±0.67** | **55.23±0.33** | **99.78±0.22** | 99.65±0.30 | **76.02±0.41** | **76.11±0.23** |
| TCL | (R4Q2)PGD | 9.26±0.25 | 13.82±0.29 | 9.35±0.47 | 16.52±0.12 | 23.26±0.24 | 38.42±0.45 | 16.26±0.19 | 24.51±0.09 | 18.09±0.16 | 24.49±0.67 | 94.01±0.17 | 96.83±0.09 |
| | (R4Q2)BERT-Attack | 29.94±0.00 | 39.21±0.00 | 25.64±0.00 | 39.56±0.00 | 41.62±0.00 | 59.57±0.00 | 36.14±0.00 | 48.78±0.00 | 9.91±0.00 | 23.64±0.00 | 37.09±0.00 | 53.07±0.00 |
| | (R4Q2)Co-Attack | 32.20±0.22 | 41.78±0.17 | 27.28±0.09 | 42.20±0.22 | 45.89±0.37 | 61.21±0.30 | 40.33±0.27 | 50.54±0.14 | 45.09±0.19 | 58.12±0.38 | 94.17±0.15 | 96.80±0.04 |
| | (R4Q2)SGA | 39.32±0.24 | 47.66±0.16 | 29.92±0.02 | 44.79±0.13 | 48.38±0.09 | 60.88±0.25 | 42.62±0.19 | 52.02±0.31 | 68.90±0.27 | 74.51±0.11 | 99.15±0.05 | 99.96±0.02 |
| | (R4Q2)ETU | 11.59±0.55 | 16.27±0.32 | 24.45±0.27 | 35.89±0.22 | 20.06±0.08 | 27.54±0.12 | 19.78±0.26 | 27.46±0.11 | 20.18±0.17 | 23.65±0.22 | 91.06±0.07 | 89.40±0.05 |
| | (R4Q2)VLATTACK | 36.52±0.67 | 21.22±0.23 | 31.47±0.20 | 45.75±0.19 | 43.88±0.11 | 54.92±0.04 | 36.83±0.17 | 47.78±0.26 | 34.26±0.13 | 46.68±0.34 | 94.17±0.06 | 96.81±0.12 |
| | (R4Q2)VLPTransferAttack | 41.23±0.34 | 48.88±0.27 | 32.39±0.29 | 47.98±0.18 | 52.49±0.06 | 62.95±0.09 | 47.13±0.12 | 57.14±0.24 | 71.66±0.06 | 75.29±0.10 | **99.89±0.02** | 99.70±0.11 |
| | (R4Q2)MAA | **42.01±0.08** | **50.98±0.31** | **34.36±0.44** | **50.00±0.25** | **55.36±0.28** | **64.27±0.33** | **49.65±0.58** | **57.85±0.33** | **72.85±0.21** | **75.69±0.18** | 99.89±0.11 | **99.95±0.05** |

$224 \times 224$ to $384 \times 384$ to attack ALBEF and TCL, or vice versa. To observe the performance trend of the proposed MMA compared to baseline methods across various retrieval stages, we engage evaluation metrics R@1, R@5, and R@10. Two benchmark datasets are utilized, and the CLIP$_{ViT-B/16}$ model is employed as the source model for training. The corresponding results are reported in Table 2. To provide a comprehensive overview of the performance gap between the baseline methods and MMA, Table 3 showcases the effectiveness of attacks based on all combinations of source-target model pairs using different attack generation methods. Given space constraints, we focus on presenting R@1 results for one of the benchmark datasets to highlight key trends succinctly.

It is evident that the compared methods experience significant performance degradation when attacking unseen models, highlighting their tendency to overfit. Specifically, adversarial examples perform better when the source and target models share similar objectives, training schemes and/or architectures, such as between CLIP models, compared to dissimilar ones, like between CLIP and other models. This discrepancy arises because different objectives, training schemes, and architectures lead to model-specific multimodal data processing. As a result, attackers depend on model-specific features

Table 4: Performance on visual grounding under different attacks on RefCOCO+. CLIP$_{\text{ViT-B/16}}$ and ALBEF for image-text retrieval serve as the source model, while ALBEF built for visual grounding is used as the target model. "Baseline" refers to the target model's performance on clean data. Smaller values indicate better adversarial transferability, and **bold** highlights the best results.

| Source Model | CLIP$_{\text{ViT-B/16}}$ | | | (R2Q1,R3W1) ALBEF | | |
|---|---|---|---|---|---|---|
| | Val | TestA | TestB | Val | TestA | TestB |
| (R2Q1,R3W1)Baseline | $51.42 \pm 0.33$ | $56.85 \pm 0.27$ | $44.77 \pm 0.19$ | $51.42 \pm 0.33$ | $56.85 \pm 0.27$ | $44.77 \pm 0.19$ |
| (R2Q1,R3W1)PGD | $51.00 \pm 0.07$ | $56.49 \pm 0.09$ | $44.68 \pm 0.10$ | $49.92 \pm 0.08$ | $56.27 \pm 0.15$ | $40.89 \pm 0.71$ |
| (R2Q1,R3W1)BERT-Attack | $43.08 \pm 0.00$ | $48.31 \pm 0.00$ | $37.23 \pm 0.00$ | $46.00 \pm 0.00$ | $52.03 \pm 0.00$ | $35.84 \pm 0.00$ |
| (R2Q1,R3W1)Co-Attack | $43.45 \pm 0.13$ | $47.80 \pm 0.27$ | $37.43 \pm 0.35$ | $43.22 \pm 0.48$ | $47.80 \pm 0.12$ | $35.08 \pm 0.34$ |
| (R2Q1,R3W1)SGA | $45.24 \pm 0.31$ | $49.79 \pm 0.23$ | $38.82 \pm 0.09$ | $44.94 \pm 0.37$ | $49.81 \pm 0.23$ | $36.61 \pm 0.26$ |
| (R2Q1,R3W1)ETU | $50.36 \pm 0.34$ | $55.24 \pm 0.24$ | $43.52 \pm 0.11$ | $48.32 \pm 0.17$ | $52.64 \pm 0.09$ | $40.01 \pm 0.23$ |
| (R2Q1,R3W1)VLATTACK | $46.32 \pm 0.38$ | $50.44 \pm 0.56$ | $39.62 \pm 0.27$ | $42.38 \pm 0.26$ | $47.48 \pm 0.31$ | $35.52 \pm 0.19$ |
| (R2Q1,R3W1,R5Q1)VLPTransferAttack | $44.69 \pm 0.22$ | $49.72 \pm 0.24$ | $38.49 \pm 0.37$ | $42.76 \pm 0.33$ | $48.09 \pm 0.51$ | $35.45 \pm 0.27$ |
| (R2Q1,R3W1)MAA | $\mathbf{41.23 \pm 0.14}$ | $\mathbf{45.32 \pm 0.23}$ | $\mathbf{35.92 \pm 0.15}$ | $\mathbf{41.35 \pm 0.20}$ | $\mathbf{46.70 \pm 0.17}$ | $\mathbf{33.40 \pm 0.33}$ |

Table 5: Performance in the image captioning task under various attacks on MSCOCO. CLIP$_{\text{ViT-B/16}}$ and ALBEF for image-text retrieval serve as the source model, while BLIP built for image captioning is used as the target model. "Baseline" refers to the target model's performance on clean data. Smaller values indicate better adversarial transferability, and **bold** highlights the best results.

| Source Model | CLIP$_{\text{ViT-B/16}}$ | | | | | (R2Q1,R3W1)ALBEF | | | | |
|---|---|---|---|---|---|---|---|---|---|---|
| | B@4 | METEOR | ROUGE_L | CIDEr | SPICE | B@4 | METEOR | ROUGE_L | CIDEr | SPICE |
| (R2Q1,R3W1)Baseline | $39.59 \pm 0.44$ | $30.87 \pm 0.16$ | $59.67 \pm 0.09$ | $132.02 \pm 0.82$ | $23.51 \pm 0.15$ | $39.59 \pm 0.44$ | $30.87 \pm 0.16$ | $59.67 \pm 0.09$ | $132.02 \pm 0.82$ | $23.51 \pm 0.15$ |
| (R2Q1,R3W1)Co-Attack | $38.34 \pm 0.06$ | $30.06 \pm 0.21$ | $58.89 \pm 0.14$ | $128.16 \pm 0.31$ | $23.12 \pm 0.26$ | $34.12 \pm 0.14$ | $27.96 \pm 0.22$ | $55.88 \pm 0.51$ | $112.73 \pm 0.25$ | $20.81 \pm 0.17$ |
| (R2Q1,R3W1)SGA | $37.19 \pm 0.24$ | $29.70 \pm 0.11$ | $58.15 \pm 0.16$ | $124.36 \pm 0.32$ | $22.27 \pm 0.46$ | $38.51 \pm 0.23$ | $30.24 \pm 0.15$ | $59.05 \pm 0.27$ | $128.99 \pm 0.41$ | $23.18 \pm 0.08$ |
| (R2Q1,R3W1)ETU | $38.92 \pm 0.44$ | $30.35 \pm 0.23$ | $59.24 \pm 0.19$ | $130.01 \pm 0.25$ | $23.34 \pm 0.09$ | $34.70 \pm 0.28$ | $27.66 \pm 0.47$ | $55.63 \pm 0.16$ | $128.57 \pm 0.24$ | $22.21 \pm 0.15$ |
| (R2Q1,R3W1)VLATTACK | $38.03 \pm 0.36$ | $30.07 \pm 0.19$ | $58.95 \pm 0.19$ | $127.64 \pm 0.23$ | $22.90 \pm 0.09$ | $34.74 \pm 0.21$ | $28.24 \pm 0.07$ | $56.19 \pm 0.23$ | $114.12 \pm 0.27$ | $21.09 \pm 0.19$ |
| (R2Q1,R3W1,R5Q1)VLPTransferAttack | $36.15 \pm 0.18$ | $29.01 \pm 0.32$ | $57.26 \pm 0.28$ | $120.58 \pm 0.12$ | $21.93 \pm 0.26$ | $28.63 \pm 0.16$ | $25.05 \pm 0.18$ | $51.73 \pm 0.52$ | $92.59 \pm 0.11$ | $18.34 \pm 0.37$ |
| (R2Q1,R3W1)MAA | $\mathbf{33.26 \pm 0.41}$ | $\mathbf{26.78 \pm 0.39}$ | $\mathbf{54.68 \pm 0.17}$ | $\mathbf{107.51 \pm 0.65}$ | $\mathbf{19.83 \pm 0.12}$ | $\mathbf{22.82 \pm 0.35}$ | $\mathbf{21.88 \pm 0.28}$ | $\mathbf{47.85 \pm 0.29}$ | $\mathbf{84.03 \pm 0.19}$ | $\mathbf{15.91 \pm 0.24}$ |

to generate adversarial examples, resulting in limited abilities to disturb cross-modal relationships. In contrast, MAA explores the representative and intrinsic characteristics and vulnerabilities of original images, thereby reducing over-reliance on source models and fostering the development of more model-generic adversarial examples. Second, BERT-Attack demonstrates less overfitting to source models. This is likely because most VLP models share the same text encoder. So all compared multi-modal attack methods utilize BERT-Attack to enhance adversarial transferability. However, as shown in Table 1, when text perturbations are removed, these methods experience a notable performance drop. Additionally, text perturbations typically involve replacing, removing, or transforming keywords, tend to be conspicuous and easily detectable Li et al. (2020); Jin et al. (2020); Iyyer et al. (2018); Naik et al. (2018); Ren et al. (2019). These highlight the importance of enhancing attacks in the image modality. Third, ETU is a universal attack method that learns uniform adversarial perturbations for all data, making it independent of sample-specific characteristics and vulnerabilities. As a result, ETU achieves inferior performance compared to sample-specific methods.

### 3.2.2 RESULTS ON THE VISUAL GROUNDING AND IMAGE CAPTIONING

The performance of various attacks on the visual grounding and image captioning tasks is presented in Tables 4 and 5. Notably, the image captioning task only utilizes images as input. Since Co-Attack reduces to the PGD attack when text perturbations are omitted, we do not report PGD results separately. Visual Grounding aims to align textual descriptions with relevant objects or locations within the visual input. This task requires attackers to effectively disrupt the correlation between texts and fine-grained image contents. (R5W1)By means of RScrop and MGSD, the proposed method can find more fine-grained characteristics and vulnerabilities of data and break the intra- and inter-model relationships across different granularities and hierarchical levels, it achieves better performance than other methods. In the image captioning task, the goal is to generate descriptive textual information that accurately reflects the content of an image. This requires models to effectively identify objects, attributes, and contextual information. Through the sliding operation of RScrop, MAA effectively considers local regions and their dependencies to generate adversarial examples, thus enhancing the ability to prevent target models from recognizing visual elements. (R5W1)In addition, as the proposed method pays attention to the characteristics of each sample, it can prevent adversarial example generation from depending on model-specific features, further promoting transferability.

In summary, our MAA takes into account both global and fine-grained information, achieving superior performance across various tasks, whether in image-text retrieval that focuses on matching entire images to text or in visual grounding and image captioning that emphasize fine-grained content.

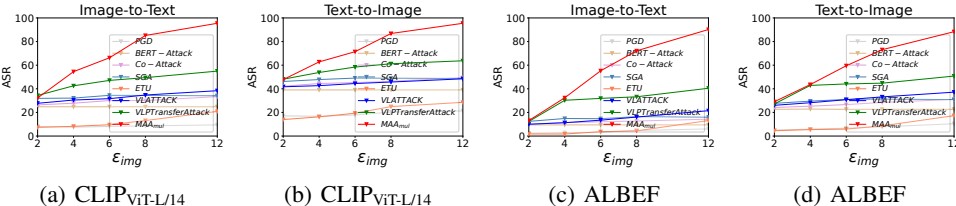

(a) CLIP$_{\text{ViT-L/14}}$      (b) CLIP$_{\text{ViT-L/14}}$      (c) ALBEF      (d) ALBEF

Figure 2: Test accuracy on Flickr30K with varying image perturbation magnitudes. The source model is CLIP$_{\text{ViT-B/16}}$. The attack success rate (ASR, %) of R@1 is reported.

### 3.3 PARAMETER ANALYSIS

#### 3.3.1 EFFECT OF VARYING PERTURBATION MAGNITUDES

As the proposed method focuses primarily on images and all multi-modal attack methods all use the same text attack techniques, we evaluate the impact of perturbation magnitude on adversarial images by fixing the text perturbation. The results are shown in Figure 2. As illustrated, our proposed method consistently outperforms others across various perturbation magnitudes. Moreover, as the perturbation magnitude increases, the performance improvement of our method becomes significantly more pronounced, while other methods show only minor gains. The reason is that the proposed method explores local regions and detailed information to uncover the intrinsic vulnerabilities of original data, which encourages model-generic adversarial examples and reduces overreliance on specific models. With the perturbation magnitude increasing, the model-generic aspects of adversarial examples would be further enhanced, thus promoting transferability. In contrast, the compared methods primarily rely on model-specific features, meaning that increased perturbation magnitudes mainly enhance model-specific adversarial examples, leading to inferior transferability.

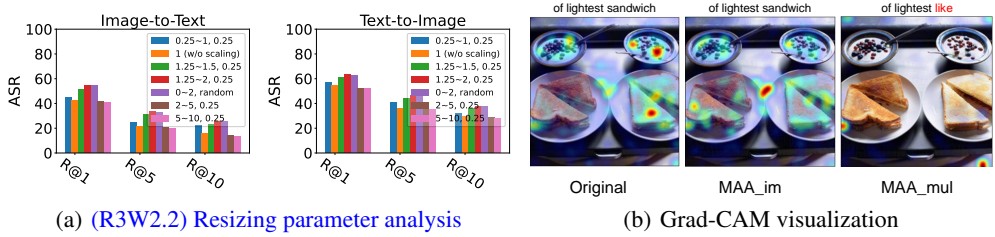

(a) (R3W2.2) Resizing parameter analysis      (b) Grad-CAM visualization

Figure 3: (a) (R3W2.2) Resizing parameter analysis: appropriate scaling can achieve competitive performance. (b) A Grad-CAM visualization of original data pairs, image-only perturbed pairs and multi-modal perturbed pairs, where MAA can significantly shift the attention of target models.

#### 3.3.2 EFFECT OF RESIZING FACTORS

Setting appropriate resizing factors is crucial for effectively exploring local details. (R3W2.2) To verify this, We evaluate MAA across various resizing ranges with intervals of 0.25, and also report the results of random selection, as shown in Figure 3(a). It is evident that scaling within an appropriate range enhances performance while exceeding this range adversely affects it. In light of scale-invariant properties of DNNs Lin et al. (2019), RScrop resizes images to attend to subtle variations in local features and ensure that the network can focus on finer details. However, as the resizing factor increases and images become progressively larger, the network may encounter diminishing returns in its ability to capture meaningful details. This assumption is supported by a consistent performance improvement within the range of 1 to 2, with peak effectiveness observed between 1.25 and 2. However, a performance decline becomes apparent once the resizing factor exceeds 2. We interpret this observation as that adversarial examples generated on well-explored features and details are more likely to transfer to other models. However, when images are excessively enlarged, critical features may become blurred or lose detail, making recognition difficult for source models. Conse-

Table 6: Ablation study of different components on Flickr30K. The attack success rate of R@1 on image-text retrieval is reported. $\text{CLIP}_{\text{ViT-B/16}}$ is adopted as the source model.

| Target Model | CLIP | | | | | | ALBEF | | TCL | |
| --- | --- | --- | --- | --- | --- | --- | --- | --- | --- | --- |
| | ViT-B/16 | | ViT-L/14 | | RN101 | | | | | |
| Method | I2T | T2I | I2T | T2I | I2T | T2I | I2T | T2I | I2T | T2I |
| (R3W2.1,R5Q1,R5W1)MAA w DIM | 99.63 | 99.87 | 42.12 | 53.09 | 62.45 | 65.13 | 27.22 | 41.3 | 31.61 | 44.5 |
| (R3W2.1,R5Q1,R5W1)MAA w TI-DIM | 96.13 | 99.27 | 32.02 | 46.3 | 41.89 | 53.14 | 24.81 | 31.62 | 24.75 | 29.6 |
| (R3W2.1,R5Q1,R5W1)MAA w SI-NI-TI-DIM | 99.85 | 99.19 | 26.99 | 41.59 | 34.11 | 44.91 | 28.95 | 33.51 | 22.64 | 29.98 |
| (R3W2.1,R5Q1,R5W1)MAA w SIA | 99.51 | 99.23 | 42.94 | 54.25 | 63.22 | 69.19 | 27.42 | 40.79 | 35.09 | 45.31 |
| MAA w ScMix | 99.75 | 99.84 | 24.66 | 42.30 | 37.04 | 49.31 | 12.30 | 27.46 | 13.28 | 28.74 |
| MAA w/o Resizing | **100.00** | 99.94 | 42.58 | 54.41 | 62.32 | 68.54 | 22.94 | 37.93 | 30.03 | 42.55 |
| MAA w/o Sliding | 99.88 | **100.00** | 26.5 | 42.01 | 38.60 | 42.92 | 12.41 | 28.11 | 14.75 | 29.43 |
| MAA w/o RScrop | **100.00** | **100.00** | 25.47 | 41.85 | 38.83 | 49.23 | 12.62 | 27.48 | 13.91 | 28.90 |
| MAA w/o MGSD | 99.51 | 99.71 | 45.02 | 56.55 | 57.81 | 65.82 | 27.63 | 43.59 | 32.77 | 45.48 |
| MAA | 99.75 | 99.05 | **54.36** | **63.02** | **72.23** | **74.87** | **32.45** | **43.62** | **41.42** | **50.29** |

quently, adversarial examples constructed from such indistinguishable features are less effective in attacking target models. (R3W2.2)Notably, scaling factors less than 1 can also enhance performance as they increase data diversity to prevent overfitting. However, they do not contribute to extracting fine-grained details, resulting in inferior performance compared to our method.

## 3.4 ABLATION STUDY

To verify the effectiveness of the proposed method, we conduct an extensive ablation study. Since the method is simple, involving only two components: RScrop and MGSD, we create two variants by removing each component individually, labeled as MAA w/o RScrop and MAA w/o MGSD. Furthermore, our MGSD employs scaling to enhance the exploration of fine-grained details and utilizes sliding crops to capture local dependencies and ensure comprehensive image coverage. To further assess the impact of these operations, we construct two variants that use only scaling and sliding, denoted as MAA w/o Sliding and MAA w/o Resizing, respectively. (R3Q2.1) We also compare the proposed Rscrop with other augmentations, including DIM Xie et al. (2019), TI-DIM Dong et al. (2019), SI-NI-TI-DIM Lin et al. (2019), SIA Wang et al. (2023)and ScMix Zhang et al. (2024). The results for all variants in the context of image-text retrieval are summarized in Table 6. The results show that the two key strategies of MAA are complementary, enabling it to explore local details across different granularities and hierarchical levels. RScrop contributes more compared to MGSD, as it still helps the method capture local information by focusing on output features, without relying on intermediate features across levels. The performance of MAA w/o Sliding demonstrates that ensuring full image coverage is crucial for thoroughly exploring the characteristics and vulnerabilities of original samples. Similarly, scaling is important for capturing finer local details.

## 3.5 VISUALIZATION

A set of Grad-CAM Selvaraju et al. (2017) visualization examples is shown in Figure 3(b), which highlights the activation areas and helps understand the regions of the image that contribute most to the model's predictions. This visualization reveals distinct focus shifts, indicating that the proposed MAA method effectively modulates the attention patterns of the target models. Although the image perturbations generated by MAA remain visually imperceptible to the human eye, they are highly effective at misleading the models, verifying the distinguish performance of the proposed solution when assessing the robustness of VLP models. In addition, text perturbations tend to be more noticeable. We leave the study of improving imperceptibly of adversarial texts in our future work.

## 4 RELATED WORK

### 4.1 VISION-LANGUAGE PRE-TRAINED MODELS

Vision-language pre-trained (VLP) models play a crucial role in advancing the understanding of visual and textual information and their interactions. By leveraging large-scale unlabeled datasets and self-supervised learning techniques, VLP Models can learn rich, generalized representations of both visual and linguistic data. This transfer learning capability enables them to be fine-tuned for a wide range of tasks with relatively small datasets, including multi-modal retrieval, zero-shot learning, image captioning, visual question answering, and visual entailment Radford et al. (2021);

Li et al. (2022; 2021); Yang et al. (2022). Notable VLP methods include CLIP Radford et al. (2021), BLIP Li et al. (2022), ALBEF Li et al. (2021), and TCL Yang et al. (2022). These methods primarily leverage multi-modal contrastive learning to align image-text pairs. Specifically, CLIP employs unimodal encoders to project data from different modalities into a unified feature space. BLIP Li et al. (2022) refines noisy captions to enhance learning effectiveness. ALBEF Li et al. (2021) and TCL Yang et al. (2022) both utilize a multi-modal encoder to learn joint representations for images and texts. ALBEF Li et al. (2021) focuses on inter-modal relationships while TCL Yang et al. (2022) considers both intra- and inter-modal relationships.

## 4.2 Adversarial Attack

With the widespread adoption of Deep Neural Networks, it is essential to evaluate their robustness to ensure their reliability in real-world applications, which often involve uncertainty and potential threats. Adversarial attacks are a prominent method used to assess this robustness, which aim to mislead model predictions by introducing imperceptible perturbations into the data Szegedy et al. (2014); Zhang et al. (2023; 2021a); Madry et al. (2018); Moosavi-Dezfooli et al. (2017). Traditional methods typically focus on specific tasks and unimodal cases, such as image classification. For image attacks, most techniques learn pixel-level perturbations, whereas text attacks often involve re-placing or removing keywords, or performing text transformations Li et al. (2020); Jin et al. (2020); Iyyer et al. (2018); Naik et al. (2018); Ren et al. (2019). Recently, there has been growing interest in multi-modal vision-language scenarios. For instance, Zhang et al. (2023); Wang et al. (2021b); Zhu et al. (2023b) address image-text retrieval by increasing the embedding distance between adversarial and original data pairs. Xu et al. (2018) learn image perturbations by minimizing the distance between the output and the target label while maximizing the difference from the original label for the visual question answering task.

**Adversarial transferability.** Early attack methods typically assume a white-box setting, where all necessary information for generating adversarial examples, including target models and tasks, is available. However, in real-world scenarios, such comprehensive information is often unavailable. To address this challenge, one approach uses an ensemble of models as the victim model during training Liu et al. (2016); Dong et al. (2018; 2019); Xiong et al. (2022), based on the intuition that adversarial examples effective against a diverse set of models are likely to mislead more models. However, assembling such a model ensemble can be difficult. Another approach utilizes momentum-based methods Dong et al. (2018); Long et al. (2024); Lin et al. (2019); Inkawhich et al. (2019) to stabilize gradient updates and avoid poor local maxima, though this may also divert per-turbations from effective paths. Data augmentation Xie et al. (2019); Fang et al. (2022); Wei et al. (2023); Wang et al. (2024; 2021a) increases data diversity, helping to prevent overfitting to specific models. Transferability is particularly challenging for attacks on VLP models, due to the unpredictable fine-tuning process. To enhance transferability, Zhang et al. (2022); Lu et al. (2023); Zhang et al. (2024) suggests increasing the gap between adversarial data and original image-text pairs. Some of these methods also focus on generating diverse image-text pairs through scale-invariant transformations Lu et al. (2023) and ScMix augmentations Zhang et al. (2024). Additionally, recent approaches consider the local utility of adversarial examples Yin et al. (2023) or perturbations Zhang et al. (2024). However, Yin et al. (2023) enlarges block-wise similarity between samples, which is constrained by the block size, failing to comprehensively capture local regions and their interactions. Zhang et al. (2024) does not consider the characteristics and vulnerabilities of the original sample. As a result, these methods cannot ensure the effectiveness and transferability of adversarial attacks.

## 5 Conclusions

In this paper, we propose a novel Meticulous Adversarial Attack (MAA), which demonstrates exceptional transferability across various VLP models, datasets, and downstream tasks. MAA enhances adversarial transferability by refining adversarial examples specifically in the image modality. The method is simple and easy to implement, consisting of a resizing and sliding crop technique and a multi-granularity similarity disruption strategy. Two components work synergistically to explore the representative, fine-grained characteristics and vulnerabilities of individual images, reducing over-reliance on model-specific patterns. Extensive experiments validate the effectiveness and transferability of MAA, showing that it achieves highly competitive performance.

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

Table 7: (R1W3, R5W1) The attack success rate (%, R@1) of the proposed method that utilizes different ensemble source models in image-text retrieval on Flickr30K.

| Target Model | CLIP | | | | | | | | ALBEF | | TCL | |
|---|---|---|---|---|---|---|---|---|---|---|---|---|
| | ViT-B/16 | | ViT-L/14 | | RN50 | | RN101 | | | | | |
| Method | I2T | T2I | I2T | T2I | I2T | T2I | I2T | T2I | I2T | T2I | I2T | T2I |
| (R1W3)ViT-B/16 | 99.98 | 99.99 | 54.36 | 63.02 | 77.14 | 80.58 | 73.05 | 74.85 | 31.80 | 43.62 | 41.10 | 49.81 |
| (R1W3)RN101 | 36.81 | 47.97 | 30.80 | 45.26 | 70.50 | 78.77 | 98.34 | 97.63 | 13.56 | 29.3 | 17.18 | 31.62 |
| (R1W3)ALBEF | 39.14 | 49.77 | 32.39 | 47.29 | 51.09 | 60.99 | 43.68 | 54.99 | 100.00 | 99.95 | 75.87 | 76.17 |
| (R1W3)TCL | 41.96 | 50.71 | 34.60 | 50.00 | 55.30 | 64.05 | 49.30 | 57.67 | 72.78 | 75.65 | 100.00 | 100.00 |
| (R1W3)ViT-B/16 + ViT-L/14 | 99.63 | 99.87 | 99.63 | 99.87 | 62.45 | 68.13 | 27.22 | 41.3 | 40.35 | 50.72 | 49.32 | 56.52 |
| (R1W3)RN50 + RN101 | 38.9 | 50.35 | 33.13 | 46.91 | 99.87 | 99.27 | 99.62 | 99.83 | 15.24 | 29.88 | 18.34 | 34.52 |
| (R1W3)ALBEF + TCL | 43.68 | 53.23 | 41.89 | 53.14 | 57.82 | 66.99 | 24.81 | 31.62 | 96.13 | 99.27 | 98.42 | 99.43 |
| (R1W3)ViT-B/16 + RN101 | 99.02 | 98.32 | 52.39 | 62.37 | 75.22 | 80.27 | 98.34 | 97.84 | 30.87 | 44.43 | 36.14 | 49.02 |
| (R1W3)ViT-B/16 + ALBEF | 99.75 | 99.68 | 60.74 | 69.75 | 76.63 | 79.31 | 73.56 | 75.09 | 99.58 | 99.56 | 85.04 | 84.88 |
| (R1W3)ViT-B/16 + ViT-L/14+ ALBEF | 97.67 | 98.13 | 98.65 | 97.71 | 77.39 | 78.77 | 73.95 | 77.63 | 96.77 | 96.8 | 81.66 | 81.40 |

# A EXPERIMENTS

## A.1 (R1W3) EXPERIMENTS ON ENSEMBLE MODELS.

(R1W3, R5W1) We further explore the effectiveness of the proposed attack method with various ensemble combinations of source models, including ensembles of homogeneous CLIP models, e.g., $CLIP_{RN50}$ and $CLIP_{RN101}$, heterogeneous models, e.g., $CLIP_{ViT-B/16}$ and $CLIP_{RN101}$, ensembles of models of different architectures and training process, e.g., CLIP and ALBEF. Different numbers of models are utilized including two and three. Results are summarized in Table 7. From the results, we have some key observations. On the one hand, using ensembles often improves performance, particularly when combining complementary models. For example, the combination of ViT-B/16 + ViT-L/14+ ALBEF outperforms using a single source model when attacking ALBEF and TCL. This supports our assertion in the Related Work section that methods capable of attacking a set of models are more likely to mislead more models. One the other hand, utilizing more models does not always lead to better performance. For instance, ViT-B/16 + $CLIP_{RN101}$ performs worse than using ViT-B/16 alone. This may be due to the varying contributions of different models to the attack. In this case, ViT-B/16 is generally more effective than $CLIP_{RN101}$, and the less effective model may dilute the overall performance when combined." We also discuss the limitations of this approach: "However, there are two main limitations to such a method. First, in many scenarios, diverse models might not be readily available. Second, utilizing multiple models significantly increases computational overhead, which can hinder practicability and scalability.

## A.2 (R1W2) FINE-GRAINED INFORMATION CAPTURE MEASUREMENT.

(R1W2) To demonstrate that the proposed method can capture fine-grained information, we perform experiments in visual grounding. Results are summarized in Table 8. The evaluation metric measures the possibility of the overlap between the predicted and ground-truth regions. If the overlap is larger than a certain value (i.e., 0.5), consider they are the same. From the result, it can be observed that the proposed method can better extract overlapped regions with the ground truth, which means that more regions are covered and contextual information is captured. We further utilize the Grad-CAM visualization method to highlight the activation maps of images, which indicates the contribution of regions to the model's predictions. Some examples are shown in Figure 4. From the visualizations, it can be observed that the proposed method captures finer-grained contextual features more effectively than the vanilla model. First, the proposed method can help focus on image regions relevant to the caption. For example, it captures the clothes in image (a), the area contributing to the bending behavior in image (c), and the entirety of the motorcycle in image (d). Second, the method can help identify cohesive areas, such as the faces and clothing of the man in image (b), the cat in image (e), and the pitcher in image (g). This demonstrates the improved contextual awareness of our approach. Third, in image (d), the proposed method captures features for both motorcycles, reflecting an improved understanding of relationships between objects described in the caption. Lastly, the proposed method emphasizes the entire area of relevant objects rather than isolating specific regions. For example, in image (a), it highlights multiple areas of the bed instead of concentrating on just a few regions. These results highlight that the proposed method effectively

Table 8: (R1W2) Performance on visual grounding of baseline model ALBEF with and without the proposed method.

| Source Model | CLIPViT-B/16 | | |
|---|---|---|---|
| | Val | TestA | TestB |
| (R2Q1)Model w/o the proposed method | 51.24 | 56.71 | 44.79 |
| (R2Q1)Model w the proposed method | 56.67 | 65.37 | 45.28 |

extracts fine-grained details and contextual information, thereby enabling a deeper exploration of the characteristics and vulnerabilities of samples, which ultimately enhances attack performance.

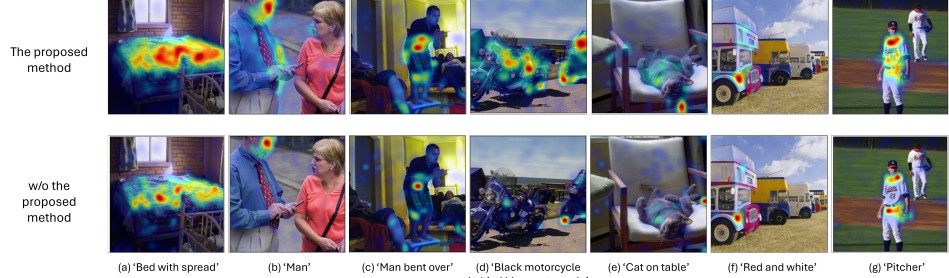

Figure 4: (R1W2) A Grad-CAM visualization of samples with and without the proposed method.

### A.3 RESULTS ON THE BLACK-BOX LARGE VISION-LANGUAGE MODELS

(R1W3,R2Q2,R5Q2) To test whether the attack methods can fool black-box Large Vision-Language Models (LVLM), we selected two open-source large vision-language models: MiniGPT-4 Zhu et al. (2023a) and Llama 3.2 Touvron et al. (2023) to ensure reproducibility of our experiments. Specifically, MiniGPT-4, based on Vicuna V0 13B (a 13-billion-parameter large language model), has recently scaled up the capabilities of large language models and demonstrates performance comparable to GPT-4. For Llama 3.2, we utilized Llama-3.2-11B-Vision-Instruct, which comprises 11 billion parameters. We evaluated the robustness of these models on multimodal tasks, including image captioning and visual question answering (VQA). For image captioning, we provided images with the prompt: "Describe this image, bringing all the details" as the input, For VQA, we input images along with the question: "What is the content of this image?" We collected the generated descriptions and answers for each attack method. To assess attack performance, we used the CLIP score Zhao et al. (2023), which measures the similarity between the features of descriptions/answers for adversarial images and those for clean images, generated by the CLIP text encoder. To ensure fair comparisons, we calculated the CLIP score between the description/answer features of adversarial images (for each attack method) and the clean image features generated by all attack methods. This is because at different query times, large vision language models would produce different responses for the same input. Additionally, we reported the CLIP score between the features of clean images and their randomly shuffled counterparts as a baseline to compare attack effectiveness. The results are summarized in Table 9. From the results, it can be observed that the proposed method can achieve the best performance in two tasks. In addition, the attack performance is not significant due to the significant gap between the source and target models, in terms of architectures, training data and schemes. Improving attack performance on large models remains an open challenge, which we identify as a direction for future research.

### A.4 COMPARISON WITH DIFFERENT AUGMENTATIONS

(R3W2.1, R3W3) To thoroughly evaluate the effectiveness of the proposed method, we compare RScrop with several popular data augmentation techniques, both with and without MGSD, including DIM Xie et al. (2019), TI-DIM Dong et al. (2019), SI-NI-TI-DIM Lin et al. (2019), SIA Wang et al. (2023), and ScMix Zhang et al. (2024). The results, presented in Table 10, reveal several insights. First, data augmentations generally enhance the transferability of adversarial examples by increasing data diversity, which reduces overfitting to source models. Second, their combinations

Table 9: (R1W3,R2Q2,R2Q1,R5Q2) Black-box attack against Large Vision-Language Models (LVLM), i.e., MiniGPT-4 and Llama 3.2, on Flickr30K in vision question answering and image captioning. $CLIP_{ViT-B/16}$ is taken as the source model. The CLIP score is used as the evaluation metric, which measures the distance between features of generated texts and references, extracted by the CLIP text encoder. For a fair comparison, we compare generated texts of adversarial images with those of clean images from all attack methods to avoid variance during different LVLM query processes. The CLIP score between generated texts for clean images and the randomly shuffled texts is taken as a bar for evaluating the attack performance. Bold indicates the best performance.

| Target Model and Task | Llama 3.2, Visual Question Answering | | | | | | MiniGPT-4, Image Captioning | | | | | |
|---|---|---|---|---|---|---|---|---|---|---|---|---|
| | CLIP Text Encoder for generating features | | | | | | CLIP Text Encoder for generating features | | | | | |
| | $CLIP_{ViT-B/16}$ | $CLIP_{ViT-B/32}$ | ViT-L/14 | RN50 | RN101 | Avg. | $CLIP_{ViT-B/16}$ | $CLIP_{ViT-B/32}$ | ViT-L/14 | RN50 | RN101 | Avg. |
| (R1W3,R2Q1,R2Q2,R5Q2)Random Shuffle | 0.415 | 0.417 | 0.312 | 0.380 | 0.507 | 0.406 | 0.445 | 0.427 | 0.302 | 0.3987 | 0.631 | 0.441 |
| (R1W3,R2Q1,R2Q2,R5Q2)Co-Attack | 0.817 | 0.814 | 0.784 | 0.802 | 0.840 | 0.811 | 0.856 | 0.823 | 0.771 | 0.847 | 0.875 | 0.834 |
| (R1W3,R2Q1,R2Q2,R5Q2)SGA | 0.815 | 0.811 | 0.783 | 0.801 | 0.839 | 0.810 | 0.862 | 0.818 | 0.775 | 0.842 | 0.869 | 0.834 |
| (R1W3,R2Q1,R2Q2,R5Q2)ETU | 0.823 | 0.820 | 0.792 | 0.810 | 0.845 | 0.818 | 0.880 | 0.825 | 0.792 | 0.862 | 0.883 | 0.848 |
| (R1W3,R2Q1,R2Q2,R5Q2)VLATTACK | 0.812 | 0.809 | 0.781 | 0.799 | 0.838 | 0.808 | 0.853 | 0.815 | 0.780 | 0.843 | 0.874 | 0.833 |
| (R1W3,R2Q1,R2Q2,R5Q2)VLPTransferAttack | 0.810 | 0.806 | 0.776 | 0.794 | 0.835 | 0.804 | 0.852 | 0.811 | 0.776 | 0.842 | 0.870 | 0.830 |
| (R1W3,R2Q1,R2Q2,R5Q2)MAA | **0.798** | **0.794** | **0.763** | **0.782** | **0.825** | **0.793** | **0.843** | **0.801** | **0.768** | **0.833** | **0.859** | **0.821** |

Table 10: (R3W2.1,R3W3,R5Q1,R5W1) Comparison with different augmentations on Flickr30K. The attack success rate of R@1 on image-text retrieval is reported. $CLIP_{ViT-B/16}$ is adopted as the source model.

| Target Model | CLIP | | | | | | ALBEF | | TCL | |
|---|---|---|---|---|---|---|---|---|---|---|
| | ViT-B/16 | | ViT-L/14 | | RN101 | | | | | |
| Method | I2T | T2I | I2T | T2I | I2T | T2I | I2T | T2I | I2T | T2I |
| (R3W2.1,R3W3,R5Q1,R5W1)MAA w DIM | 99.63 | 99.87 | 42.12 | 53.09 | 62.45 | 65.13 | 27.22 | 41.3 | 31.61 | 44.5 |
| (R3W2.1,R3W3,R5Q1,R5W1)MAA w TI-DIM | 96.13 | 99.27 | 32.02 | 46.3 | 41.89 | 53.14 | 24.81 | 31.62 | 24.75 | 29.6 |
| (R3W2.1,R3W3,R5Q1,R5W1)MAA w SI-NI-TI-DIM | 99.85 | 99.19 | 26.99 | 41.59 | 34.11 | 44.91 | 28.95 | 33.51 | 22.64 | 29.98 |
| (R3W2.1,R3W3,R5Q1,R5W1)MAA w SIA | 99.51 | 99.23 | 42.94 | 54.25 | 63.22 | 69.19 | 27.42 | 40.79 | 35.09 | 45.31 |
| (R3W2.1,R3W3,R5Q1,R5W1)MAA w DIM w/o MGSD | **100.00** | **100.00** | 41.37 | 54.72 | 62.96 | 63.48 | 26.13 | 40.5 | 29.98 | 42.30 |
| (R3W2.1,R3W3,R5Q1,R5W1)MAA w TI-DIM w/o MGSD | 99.14 | 99.36 | 38.53 | 52.35 | 44.83 | 56.4 | 25.03 | 30.64 | 23.12 | 27.23 |
| (R3W2.1,R3W3,R5Q1,R5W1)MAA w SI-NI-TI-DIM w/o MGSD | 99.94 | 99.57 | 27.48 | 42.11 | 34.87 | 44.01 | 27.48 | 33.97 | 22.14 | 28.42 |
| (R3W2.1,R3W3,R5Q1,R5W1)MAA w SIA w/o MGSD | 98.89 | 99.22 | 41.11 | 53.56 | 61.66 | 68.81 | 25.75 | 40.74 | 33.24 | 42.80 |
| MAA w ScMix | 99.75 | 99.84 | 24.66 | 42.30 | 37.04 | 49.31 | 12.30 | 27.46 | 13.28 | 28.74 |
| MAA w/o Resizing | **100.00** | 99.94 | 42.58 | 54.41 | 62.32 | 68.54 | 22.94 | 37.93 | 30.03 | 42.55 |
| MAA w/o Sliding | 99.88 | **100.00** | 26.5 | 42.01 | 38.60 | 42.92 | 12.41 | 28.11 | 14.75 | 29.43 |
| MAA w/o RScrop | **100.00** | **100.00** | 25.47 | 41.85 | 38.83 | 49.23 | 12.62 | 27.48 | 13.91 | 28.90 |
| MAA w/o MGSD | 99.51 | 99.71 | 45.02 | 56.55 | 57.81 | 65.82 | 27.63 | 43.59 | 32.77 | 45.48 |
| MAA | 99.75 | 99.05 | **54.36** | **63.02** | **72.23** | **74.87** | **32.45** | **43.62** | **41.42** | **50.29** |

with MGSD provide only minor performance improvements, likely due to their limited ability to alleviate fixed-size processing constraints in models, which restrict the extraction of fine-grained details. In contrast, through resizing and sliding crop, the proposed RScrop effectively addresses this constraint, capturing finer-grained information that shifts adversarial example generation to rely more on sample-specific characteristics rather than model-specific features. These findings highlight the superior ability of the proposed method to extract detailed information and improve adversarial transferability.

## A.5 RESULTS ON MORE MODELS

Table 11: The attack success rate $(\%)$ on the image-text retrieval task. CLIP$_{\text{ViT-B/16}}$ is adopted as the source model for training, while the target models include CLIP$_{\text{ViT-B/16}}$, CLIP$_{\text{ViT-L/14}}$, CLIP$_{\text{ResNet50}}$, CLIP$_{\text{ResNet101}}$, ALBEF, and TCL. The grey background indicates the white-box attack results. **Bold** indicates the best results.

| Dataset | | Flickr30K | | | | | | MSCOCO | | | | | |
|---|---|---|---|---|---|---|---|---|---|---|---|---|---|
| | | Image-to-Text | | | Text-to-Image | | | Image-to-Text | | | Text-to-Image | | |
| Target Model | Method | R@1 | R@5 | R@10 | R@1 | R@5 | R@10 | R@1 | R@5 | R@10 | R@1 | R@5 | R@10 |
| CLIP$_{\text{ViT-B/16}}$ | (R4Q2)PGD | 90.55 ± 0.89 | 78.09 ± 0.52 | 78.42 ± 0.34 | 98.68 ± 0.82 | 95.83 ± 0.68 | 94.23 ± 0.56 | 97.44 ± 0.11 | 94.53 ± 0.24 | 91.06 ± 0.23 | 96.53 ± 0.26 | 91.90 ± 0.33 | 88.57 ± 0.49 |
| | (R4Q2)BERT-Attack | 28.34 ± 0.00 | 11.63 ± 0.00 | 6.71 ± 0.00 | 39.05 ± 0.00 | 24.06 ± 0.00 | 17.40 ± 0.00 | 55.25 ± 0.00 | 37.26 ± 0.00 | 28.93 ± 0.00 | 57.86 ± 0.00 | 45.05 ± 0.00 | 38.73 ± 0.00 |
| | (R4Q2)Co-Attack | 97.73 ± 0.18 | 94.29 ± 0.02 | 92.08 ± 0.40 | 98.83 ± 0.05 | 96.17 ± 0.05 | 94.38 ± 0.08 | 99.50 ± 0.12 | 98.75 ± 0.10 | 97.86 ± 0.35 | 99.60 ± 0.06 | 98.89 ± 0.05 | 98.25 ± 0.12 |
| | (R4Q2)SGA | 99.53 ± 0.05 | 97.96 ± 0.03 | 96.48 ± 0.08 | 99.73 ± 0.03 | 98.88 ± 0.03 | 97.72 ± 0.12 | 91.91 ± 0.02 | 82.96 ± 0.21 | 78.69 ± 0.09 | 91.18 ± 0.12 | 81.57 ± 0.32 | 79.93 ± 0.08 |
| | (R4Q2)ETU | 97.40 ± 0.09 | 94.16 ± 0.54 | 78.84 ± 0.32 | 90.24 ± 0.22 | 84.31 ± 0.26 | 69.32 ± 0.28 | 91.56 ± 0.19 | 80.45 ± 0.21 | 78.87 ± 0.37 | 90.06 ± 0.24 | 88.63 ± 0.09 | 86.17 ± 0.43 |
| | (R4Q2)VLATTACK | 99.86 ± 0.14 | **99.99 ± 0.01** | 99.50 ± 0.23 | 99.92 ± 0.05 | 99.54 ± 0.24 | **99.56 ± 0.06** | **99.93 ± 0.07** | 99.79 ± 0.11 | 99.75 ± 0.06 | **99.98 ± 0.02** | **99.96 ± 0.04** | **99.89 ± 0.06** |
| | (R5Q1)VLPTransferAttack | **99.98 ± 0.02** | 99.53 ± 0.03 | 99.42 ± 0.07 | **99.99 ± 0.01** | **99.76 ± 0.02** | 99.53 ± 0.01 | **99.93 ± 0.03** | **99.90 ± 0.01** | 99.77 ± 0.02 | 99.95 ± 0.03 | 99.88 ± 0.02 | 99.80 ± 0.02 |
| | (R4Q2)MAA | 99.75 ± 0.02 | 99.05 ± 0.02 | 98.65 ± 0.07 | 99.66 ± 0.05 | 99.25 ± 0.06 | 98.85 ± 0.04 | **99.93 ± 0.05** | 99.81 ± 0.07 | **99.92 ± 0.02** | 99.90 ± 0.05 | 99.70 ± 0.26 | 99.79 ± 0.11 |
| CLIP$_{\text{ViT-L/14}}$ | (R4Q2)PGD | 7.56 ± 0.42 | 2.02 ± 0.38 | 0.59 ± 0.18 | 16.71 ± 0.56 | 6.27 ± 0.53 | 3.66 ± 0.26 | 20.25 ± 0.52 | 10.19 ± 0.28 | 7.04 ± 0.09 | 29.10 ± 0.31 | 16.13 ± 0.24 | 12.49 ± 0.18 |
| | (R4Q2)BERT-Attack | 24.79 ± 0.00 | 9.76 ± 0.00 | 5.59 ± 0.00 | 39.05 ± 0.00 | 23.01 ± 0.00 | 17.07 ± 0.00 | 51.05 ± 0.00 | 32.89 ± 0.00 | 25.82 ± 0.00 | 59.63 ± 0.00 | 44.27 ± 0.00 | 37.56 ± 0.00 |
| | (R4Q2)Co-Attack | 27.80 ± 0.18 | 10.44 ± 0.06 | 5.29 ± 0.10 | 44.50 ± 0.64 | 27.31 ± 0.35 | 19.93 ± 0.04 | 53.58 ± 0.32 | 36.24 ± 0.26 | 28.29 ± 0.31 | 64.28 ± 0.17 | 48.89 ± 0.20 | 41.99 ± 0.15 |
| | (R4Q2)SGA | 32.14 ± 0.28 | 15.52 ± 0.12 | 9.63 ± 0.22 | 47.83 ± 0.06 | 30.21 ± 0.13 | 23.17 ± 0.22 | 57.28 ± 0.11 | 41.33 ± 0.46 | 33.28 ± 0.17 | 65.98 ± 0.21 | 50.89 ± 0.22 | 44.03 ± 0.41 |
| | (R4Q2)ETU | 8.22 ± 0.47 | 1.12 ± 0.08 | 0.20 ± 0.01 | 16.24 ± 0.32 | 5.45 ± 0.26 | 3.46 ± 0.32 | 20.22 ± 0.09 | 10.46 ± 0.15 | 7.92 ± 0.06 | 25.53 ± 0.18 | 14.58 ± 0.16 | 10.44 ± 0.20 |
| | (R4Q2)VLATTACK | 30.49 ± 0.06 | 12.54 ± 0.38 | 6.04 ± 0.09 | 42.69 ± 0.28 | 26.64 ± 0.19 | 19.32 ± 0.25 | 56.22 ± 0.23 | 37.39 ± 0.19 | 29.91 ± 0.43 | 61.55 ± 0.53 | 45.28 ± 0.37 | 37.12 ± 0.32 |
| | (R4Q2,R5Q1)VLPTransferAttack | 42.54 ± 0.52 | 23.09 ± 0.33 | 14.55 ± 0.41 | 53.82 ± 0.05 | 36.24 ± 0.19 | 28.39 ± 0.27 | 64.82 ± 0.28 | 48.98 ± 0.19 | 40.54 ± 0.06 | 71.85 ± 0.24 | 58.66 ± 0.30 | 51.77 ± 0.34 |
| | (R4Q2)MAA | **54.52 ± 0.91** | **33.36 ± 0.12** | **25.76 ± 0.16** | **62.80 ± 0.50** | **46.32 ± 0.27** | **38.11 ± 0.21** | **80.72 ± 0.13** | **67.83 ± 0.14** | **61.75 ± 0.73** | **83.42 ± 0.49** | **72.67 ± 0.36** | **67.03 ± 0.21** |
| CLIP$_{\text{ResNet50}}$ | (R4Q2)PGD | 14.89 ± 0.46 | 4.884 ± 0.51 | 2.02 ± 0.27 | 24.23 ± 0.39 | 9.05 ± 0.24 | 5.56 ± 0.16 | 29.03 ± 0.33 | 15.39 ± 0.20 | 10.88 ± 0.15 | 38.12 ± 0.56 | 22.64 ± 0.36 | 16.99 ± 0.23 |
| | (R4Q2)BERT-Attack | 35.25 ± 0.00 | 14.69 ± 0.00 | 8.03 ± 0.00 | 44.73 ± 0.00 | 27.7 ± 0.00 | 20.28 ± 0.00 | 59.05 ± 0.00 | 42.18 ± 0.00 | 32.88 ± 0.00 | 67.43 ± 0.00 | 52.64 ± 0.00 | 45.84 ± 0.00 |
| | (R4Q2)Co-Attack | 39.35 ± 0.44 | 17.98 ± 0.64 | 10.03 ± 0.19 | 50.54 ± 0.61 | 31.64 ± 0.78 | 25.21 ± 0.23 | 64.00 ± 0.33 | 45.92 ± 0.31 | 37.29 ± 0.24 | 71.95 ± 0.20 | 56.88 ± 0.24 | 49.78 ± 0.17 |
| | (R4Q2)SGA | 43.00 ± 0.71 | 22.88 ± 0.23 | 15.65 ± 0.34 | 56.78 ± 0.12 | 36.47 ± 0.24 | 28.89 ± 0.31 | 68.78 ± 0.46 | 53.01 ± 0.16 | 43.76 ± 0.28 | 75.46 ± 0.19 | 62.00 ± 0.22 | 54.58 ± 0.16 |
| | (R4Q2)ETU | 12.45 ± 0.26 | 3.46 ± 0.05 | 2.35 ± 0.43 | 42.69 ± 0.28 | 7.37 ± 0.28 | 5.06 ± 0.02 | 27.67 ± 0.37 | 15.11 ± 0.16 | 10.90 ± 0.12 | 37.12 ± 0.16 | 21.79 ± 0.17 | 16.10 ± 0.09 |
| | (R4Q2)VLATTACK | 44.32 ± 0.22 | 20.80 ± 0.08 | 14.26 ± 0.31 | 54.31 ± 0.31 | 35.07 ± 0.08 | 27.78 ± 0.26 | 70.28 ± 0.15 | 52.14 ± 0.19 | 43.52 ± 0.33 | 75.54 ± 0.47 | 60.99 ± 0.39 | 53.35 ± 0.62 |
| | (R5Q1)VLPTransferAttack | 52.65 ± 0.41 | 31.48 ± 0.24 | 22.47 ± 0.22 | 63.34 ± 0.46 | 43.52 ± 0.25 | 35.63 ± 0.32 | 76.35 ± 0.37 | 61.39 ± 0.22 | 53.23 ± 0.25 | 80.82 ± 0.36 | 68.29 ± 0.42 | 70.10 ± 0.23 |
| | (R4Q2)MAA | **77.02 ± 0.25** | **43.62 ± 0.64** | **52.85 ± 0.45** | **80.26 ± 0.43** | **65.42 ± 0.56** | **58.12 ± 0.24** | **92.28 ± 0.60** | **85.71 ± 0.82** | **80.46 ± 0.82** | **92.86 ± 0.52** | **86.32 ± 0.37** | **82.03 ± 0.29** |
| CLIP$_{\text{ResNet101}}$ | (R4Q2)PGD | 8.84 ± 0.67 | 2.03 ± 0.34 | 1.30 ± 0.27 | 12.82 ± 0.47 | 4.28 ± 0.28 | 2.68 ± 0.10 | 14.89 ± 0.09 | 7.92 ± 0.13 | 5.85 ± 0.06 | 21.07 ± 0.06 | 11.58 ± 0.32 | 8.91 ± 0.12 |
| | (R4Q2)BERT-Attack | 30.27 ± 0.00 | 11.63 ± 0.00 | 5.77 ± 0.00 | 37.39 ± 0.00 | 24.92 ± 0.00 | 18.59 ± 0.00 | 52.39 ± 0.00 | 35.55 ± 0.00 | 28.76 ± 0.00 | 58.64 ± 0.00 | 46.29 ± 0.00 | 39.39 ± 0.00 |
| | (R4Q2)Co-Attack | 35.93 ± 0.48 | 13.80 ± 0.11 | 8.75 ± 0.12 | 44.52 ± 0.45 | 29.13 ± 0.23 | 22.88 ± 0.19 | 58.87 ± 0.74 | 40.41 ± 0.54 | 32.89 ± 0.37 | 65.54 ± 0.13 | 52.18 ± 0.16 | 45.17 ± 0.14 |
| | (R4Q2)SGA | 44.01 ± 0.29 | 21.87 ± 0.31 | 14.31 ± 0.44 | 51.19 ± 0.38 | 33.25 ± 0.51 | 26.64 ± 0.33 | 61.83 ± 0.29 | 47.87 ± 0.35 | 40.35 ± 0.38 | 69.01 ± 0.44 | 56.29 ± 0.46 | 49.65 ± 0.27 |
| | (R4Q2)ETU | 7.89 ± 0.62 | 1.97 ± 0.32 | 1.15 ± 0.06 | 12.45 ± 0.14 | 4.75 ± 0.07 | 2.88 ± 0.15 | 18.24 ± 0.62 | 12.62 ± 0.24 | 8.70 ± 0.27 | 23.01 ± 0.44 | 10.15 ± 0.20 | 9.08 ± 0.09 |
| | (R4Q2)VLATTACK | 41.31 ± 0.28 | 18.42 ± 0.17 | 11.03 ± 0.25 | 48.65 ± 0.10 | 31.47 ± 0.19 | 24.65 ± 0.15 | 62.88 ± 0.09 | 46.87 ± 0.05 | 38.80 ± 0.20 | 70.34 ± 0.32 | 56.43 ± 0.31 | 49.45 ± 0.25 |
| | (R5Q1)VLPTransferAttack | 51.64 ± 0.56 | 28.45 ± 0.19 | 19.74 ± 0.32 | 59.42 ± 0.22 | 40.83 ± 0.35 | 33.21 ± 0.44 | 70.86 ± 0.42 | 57.47 ± 0.29 | 48.63 ± 0.24 | 76.49 ± 0.37 | 64.39 ± 0.17 | 57.58 ± 0.25 |
| | (R4Q2)MAA | **72.23 ± 0.82** | **55.34 ± 0.37** | **44.89 ± 0.41** | **74.87 ± 0.23** | **60.65 ± 0.42** | **53.42 ± 0.39** | **89.56 ± 0.49** | **81.35 ± 0.27** | **76.86 ± 0.38** | **90.07 ± 0.16** | **83.12 ± 0.02** | **78.66 ± 0.06** |
| ALBEF | (R4Q2)PGD | 2.87 ± 0.09 | 0.23 ± 0.05 | 0.30 ± 0.03 | 5.85 ± 0.21 | 1.83 ± 0.11 | 1.10 ± 0.08 | 8.42 ± 0.49 | 3.87 ± 0.53 | 1.87 ± 0.08 | 13.11 ± 0.36 | 5.49 ± 0.12 | 3.50 ± 0.12 |
| | (R4Q2)BERT-Attack | 9.45 ± 0.00 | 1.31 ± 0.00 | 0.45 ± 0.00 | 22.82 ± 0.00 | 11.22 ± 0.00 | 8.15 ± 0.00 | 28.32 ± 0.00 | 12.67 ± 0.00 | 8.25 ± 0.00 | 41.16 ± 0.00 | 26.17 ± 0.00 | 20.77 ± 0.00 |
| | (R4Q2)Co-Attack | 11.42 ± 0.22 | 1.92 ± 0.18 | 0.53 ± 0.08 | 25.30 ± 0.16 | 12.71 ± 0.13 | 9.33 ± 0.13 | 31.01 ± 0.44 | 14.29 ± 0.60 | 9.37 ± 0.43 | 43.34 ± 0.73 | 27.71 ± 0.61 | 22.17 ± 0.51 |
| | (R4Q2)SGA | 14.86 ± 0.05 | 3.31 ± 0.00 | 1.40 ± 0.00 | 29.56 ± 0.02 | 14.32 ± 0.24 | 10.68 ± 0.32 | 36.62 ± 0.20 | 18.93 ± 0.08 | 13.13 ± 0.24 | 46.78 ± 0.19 | 30.74 ± 0.19 | 24.05 ± 0.31 |
| | (R4Q2)ETU | 1.72 ± 0.05 | 0.30 ± 0.00 | 0.20 ± 0.00 | 5.89 ± 0.46 | 1.26 ± 0.09 | 0.88 ± 0.03 | 12.83 ± 0.47 | 6.86 ± 0.26 | 3.34 ± 0.38 | 13.16 ± 0.67 | 7.23 ± 0.37 | 4.34 ± 0.16 |
| | (R4Q2)VLATTACK | 11.29 ± 0.15 | 2.52 ± 0.10 | 1.00 ± 0.00 | 28.22 ± 0.42 | 13.78 ± 0.15 | 10.23 ± 0.11 | 34.60 ± 0.17 | 17.21 ± 0.29 | 11.25 ± 0.24 | 46.73 ± 0.33 | 31.43 ± 0.42 | 25.34 ± 0.41 |
| | (R5Q1)VLPTransferAttack | 30.28 ± 0.29 | 12.96 ± 0.16 | 6.14 ± 0.35 | 42.88 ± 0.08 | 25.45 ± 0.12 | 19.17 ± 0.09 | 52.23 ± 0.23 | 30.39 ± 0.07 | 24.74 ± 0.12 | 59.27 ± 0.24 | 44.09 ± 0.26 | 34.21 ± 0.19 |
| | (R4Q2)MAA | **32.45 ± 0.67** | **14.68 ± 0.43** | **11.20 ± 0.21** | **43.62 ± 0.55** | **25.63 ± 0.24** | **19.83 ± 0.20** | **53.82 ± 0.52** | **33.38 ± 0.23** | **26.01 ± 0.54** | **60.78 ± 0.74** | **44.35 ± 0.29** | **36.44 ± 0.25** |
| TCL | (R4Q2)PGD | 5.28 ± 0.32 | 0.32 ± 0.05 | 0.10 ± 0.02 | 8.10 ± 0.22 | 2.31 ± 0.19 | 1.39 ± 0.12 | 10.51 ± 0.27 | 4.19 ± 0.22 | 2.40 ± 0.07 | 14.62 ± 0.31 | 6.45 ± 0.22 | 4.09 ± 0.12 |
| | (R4Q2)BERT-Attack | 9.59 ± 0.00 | 2.01 ± 0.00 | 0.60 ± 0.00 | 24.05 ± 0.00 | 11.89 ± 0.00 | 8.20 ± 0.00 | 29.15 ± 0.00 | 13.45 ± 0.00 | 9.39 ± 0.00 | 41.01 ± 0.00 | 25.96 ± 0.00 | 19.93 ± 0.00 |
| | (R4Q2)Co-Attack | 12.63 ± 0.44 | 3.09 ± 0.12 | 0.84 ± 0.19 | 25.85 ± 0.56 | 13.89 ± 0.58 | 9.29 ± 0.33 | 32.35 ± 0.32 | 14.78 ± 0.51 | 10.50 ± 0.23 | 43.72 ± 0.19 | 27.77 ± 0.33 | 21.86 ± 0.12 |
| | (R4Q2)SGA | 16.26 ± 0.21 | 3.69 ± 0.12 | 1.47 ± 0.05 | 30.66 ± 0.27 | 16.05 ± 0.19 | 11.20 ± 0.15 | 37.14 ± 0.28 | 18.98 ± 0.32 | 12.87 ± 0.08 | 47.05 ± 0.16 | 31.18 ± 0.11 | 24.79 ± 0.21 |
| | (R4Q2)ETU | 11.78 ± 0.68 | 6.12 ± 0.36 | 2.06 ± 0.11 | 17.54 ± 0.43 | 8.65 ± 0.44 | 6.08 ± 0.23 | 17.36 ± 0.28 | 5.55 ± 0.34 | 3.92 ± 0.14 | 15.72 ± 0.26 | 7.26 ± 0.33 | 5.50 ± 0.21 |
| | (R4Q2)VLATTACK | 14.49 ± 0.25 | 3.71 ± 0.19 | 1.47 ± 0.12 | 30.23 ± 0.22 | 16.34 ± 0.43 | 11.22 ± 0.32 | 35.94 ± 0.18 | 18.85 ± 0.09 | 12.67 ± 0.34 | 48.39 ± 0.28 | 32.34 ± 0.15 | 25.87 ± 0.18 |
| | (R5Q1)VLPTransferAttack | 30.66 ± 0.09 | 12.79 ± 0.16 | 6.10 ± 0.12 | 42.79 ± 0.45 | 26.52 ± 0.33 | 19.20 ± 0.58 | 56.24 ± 0.28 | 37.39 ± 0.22 | 29.58 ± 0.34 | 61.58 ± 0.27 | 45.10 ± 0.03 | 36.96 ± 0.11 |
| | (R4Q2)MAA | **41.42 ± 0.48** | **22.42 ± 0.11** | **17.03 ± 0.15** | **50.29 ± 0.41** | **31.24 ± 0.25** | **24.82 ± 0.33** | **60.95 ± 0.39** | **39.96 ± 0.22** | **31.65 ± 0.36** | **62.37 ± 0.27** | **45.17 ± 0.24** | **37.46 ± 0.28** |
| (R2Q1)BLIP | (R2Q1)PGD | 2.66 ± 0.53 | 0.36 ± 0.31 | 0.13 ± 0.34 | 5.63 ± 0.17 | 2.22 ± 0.24 | 1.12 ± 0.28 | 11.27 ± 0.14 | 4.08 ± 0.32 | 1.60 ± 0.51 | 12.07 ± 0.25 | 5.32 ± 0.18 | 2.12 ± 0.07 |
| | (R2Q1)BERT-Attack | 9.36 ± 0.00 | 1.48 ± 0.00 | 0.62 ± 0.00 | 22.01 ± 0.00 | 11.23 ± 0.00 | 8.11 ± 0.00 | 28.23 ± 0.00 | 12.89 ± 0.00 | 9.42 ± 0.00 | 39.23 ± 0.00 | 24.25 ± 0.00 | 19.98 ± 0.00 |
| | (R2Q1)Co-Attack | 10.89 ± 0.49 | 2.34 ± 0.34 | 1.21 ± 0.16 | 23.49 ± 0.43 | 13.07 ± 0.45 | 9.82 ± 0.20 | 27.65 ± 0.27 | 13.34 ± 0.50 | 9.93 ± 0.48 | 41.18 ± 0.11 | 25.85 ± 0.23 | 23.03 ± 0.17 |
| | (R2Q1)SGA | 13.29 ± 0.09 | 2.15 ± 0.30 | 1.42 ± 0.09 | 27.32 ± 0.46 | 13.17 ± 0.34 | 10.69 ± 0.25 | 35.55 ± 0.48 | 13.42 ± 0.25 | 11.96 ± 0.14 | 42.89 ± 0.23 | 28.68 ± 0.16 | 23.66 ± 0.42 |
| | (R2Q1)ETU | 2.36 ± 0.22 | 2.08 ± 0.12 | 1.89 ± 0.09 | 4.38 ± 0.41 | 2.02 ± 0.18 | 1.38 ± 0.11 | 10.90 ± 0.31 | 5.85 ± 0.17 | 4.11 ± 0.23 | 11.33 ± 0.07 | 6.26 ± 0.13 | 2.31 ± 0.09 |
| | (R2Q1)VLATTACK | 12.01 ± 0.20 | 4.88 ± 0.27 | 2.03 ± 0.06 | 26.78 ± 0.15 | 12.28 ± 0.22 | 9.89 ± 0.43 | 31.39 ± 0.24 | 18.85 ± 0.17 | 15.28 ± 0.21 | 45.72 ± 0.09 | 29.78 ± 0.53 | 23.01 ± 0.28 |
| | (R2Q1,R5Q1)VLPTransferAttack | 27.82 ± 0.35 | 12.68 ± 0.21 | 5.99 ± 0.29 | 40.14 ± 0.19 | 22.35 ± 0.82 | 15.58 ± 0.23 | 51.17 ± 0.54 | 26.59 ± 0.33 | 21.07 ± 0.48 | 56.38 ± 0.11 | 45.35 ± 0.34 | 31.26 ± 0.19 |
| | (R2Q1)MAA | **29.04 ± 0.29** | **15.11 ± 0.37** | **10.97 ± 0.62** | **41.22 ± 0.31** | **24.61 ± 0.33** | **17.57 ± 0.20** | **52.88 ± 0.15** | **30.33 ± 0.34** | **22.69 ± 0.17** | **58.78 ± 0.33** | **46.05 ± 0.40** | **34.49 ± 0.14** |

Table 12: The attack success rate $(\%)$ of R@1 in image-text retrieval on Flickr30K. Grey background highlights white-box attack results, and **bold** indicates the best performance.

| Target Model | | CLIP | | | | | | | | ALBEF | | TCL | | (R2Q1)BLIP | |
|---|---|---|---|---|---|---|---|---|---|---|---|---|---|---|---|
| | | ViT-B/16 | | ViT-L/14 | | RN50 | | RN101 | | | | | | | |
| Source Model | Method | I2T | T2I | I2T | T2I | I2T | T2I | I2T | T2I | I2T | T2I | I2T | T2I | I2T | T2I |
| CLIP$_{\text{ResNet101}}$ | (R4Q2,R2Q1)PGD | 2.22 ± 0.17 | 6.73 ± 0.09 | 8.42 ± 0.12 | 15.91 ± 0.12 | 15.28 ± 0.22 | 23.76 ± 0.19 | 93.27 ± 0.06 | 96.62 ± 0.18 | 1.78 ± 0.17 | 4.45 ± 0.11 | 3.82 ± 0.15 | 6.95 ± 0.25 | 1.56 ± 0.09 | 5.32 ± 0.17 |
| | (R4Q2,R2Q1)BERT-Attack | 27.12 ± 0.00 | 37.44 ± 0.00 | 25.28 ± 0.00 | 39.85 ± 0.00 | 35.52 ± 0.00 | 46.91 ± 0.00 | 30.52 ± 0.00 | 39.97 ± 0.00 | 9.49 ± 0.00 | 23.25 ± 0.00 | 11.59 ± 0.00 | 24.36 ± 0.00 | 7.48 ± 0.00 | 21.92 ± 0.00 |
| | (R4Q2)Co-Attack | 28.67 ± 0.27 | 40.42 ± 0.33 | 25.58 ± 0.16 | 41.68 ± 0.19 | 38.62 ± 0.46 | 52.89 ± 0.18 | 98.18 ± 0.13 | 98.57 ± 0.06 | 9.39 ± 0.12 | 23.83 ± 0.24 | 11.82 ± 0.16 | 25.18 ± 0.18 | 8.08 ± 0.40 | 19.36 ± 0.73 |
| | (R4Q2,R2Q1)SGA | 32.57 ± 0.11 | 44.37 ± 0.24 | 28.32 ± 0.09 | 43.19 ± 0.06 | 49.27 ± 0.22 | 61.28 ± 0.27 | 97.85 ± 0.11 | 97.86 ± 0.09 | 11.82 ± 0.23 | 26.51 ± 0.17 | 14.69 ± 0.12 | 28.79 ± 0.27 | 9.75 ± 0.18 | 23.48 ± 0.16 |
| | (R4Q2,R2Q1)ETU | 5.28 ± 0.09 | 7.43 ± 0.17 | 7.63 ± 0.26 | 10.15 ± 0.08 | 13.64 ± 0.23 | 16.40 ± 0.17 | 88.87 ± 0.22 | 89.52 ± 0.36 | 4.31 ± 0.21 | 7.36 ± 0.08 | 5.71 ± 0.22 | 8.98 ± 0.10 | 4.02 ± 0.28 | 7.99 ± 0.07 |
| | (R4Q2,R2Q1)VLATTACK | 33.00 ± 0.31 | 43.27 ± 0.09 | 27.80 ± 0.17 | 40.72 ± 0.15 | 48.74 ± 0.08 | 59.72 ± 0.18 | 94.25 ± 0.24 | 96.82 ± 0.07 | 3.92 ± 0.09 | 9.38 ± 0.13 | 8.68 ± 0.26 | 13.91 ± 0.31 | 5.67 ± 0.59 | 11.22 ± 0.27 |
| | (R4Q2,R2Q1)VLPTransferAttack | 35.28 ± 0.45 | 46.71 ± 0.33 | 28.41 ± 0.15 | 43.65 ± 0.29 | 60.54 ± 0.32 | 69.71 ± 0.17 | 94.13 ± 0.08 | 96.78 ± 0.14 | 12.39 ± 0.34 | 27.98 ± 0.14 | 16.65 ± 0.19 | 30.07 ± 0.09 | 11.76 ± 0.09 | 28.06 ± 0.27 |
| | (R4Q2)MAA | **36.03 ± 0.81** | **47.56 ± 0.52** | **30.06 ± 0.86** | **44.80 ± 0.66** | **70.83 ± 0.45** | **78.93 ± 0.48** | **98.98 ± 0.82** | **98.82 ± 1.01** | **14.08 ± 0.52** | **29.15 ± 0.15** | **17.01 ± 0.26** | **31.94 ± 0.42** | **12.82 ± 0.26** | **29.11 ± 0.34** |
| ALBEF | (R4Q2,R2Q1)PGD | 10.45 ± 0.54 | 14.92 ± 0.24 | 16.52 ± 0.14 | 30.83 ± 0.32 | 15.50 ± 0.11 | 23.79 ± 0.37 | 22.27 ± 0.23 | 38.24 ± 0.16 | 87.12 ± 0.08 | 90.07 ± 0.10 | 25.04 ± 0.23 | 30.12 ± 0.17 | 22.04 ± 0.38 | 26.83 ± 0.14 |
| | (R4Q2,R2Q1)BERT-Attack | 25.03 ± 0.00 | 35.66 ± 0.00 | 28.88 ± 0.00 | 38.04 ± 0.00 | 37.12 ± 0.00 | 56.90 ± 0.00 | 31.42 ± 0.00 | 45.35 ± 0.00 | 32.53 ± 0.00 | 50.61 ± 0.00 | 8.22 ± 0.00 | 19.57 ± 0.00 | 7.25 ± 0.00 | 18.68 ± 0.00 |
| | (R4Q2,R2Q1)Co-Attack | 27.88 ± 0.25 | 38.52 ± 0.09 | 24.23 ± 0.61 | 38.51 ± 0.23 | 33.26 ± 0.19 | 46.42 ± 0.10 | 33.58 ± 0.42 | 46.39 ± 0.16 | 98.61 ± 0.08 | 98.66 ± 0.05 | 28.25 ± 0.15 | 42.99 ± 0.51 | 23.36 ± 0.25 | 48.59 ± 0.18 |
| | (R4Q2,R2Q1)SGA | 38.76 ± 0.27 | 47.45 ± 0.25 | 31.20 ± 0.07 | 45.65 ± 0.32 | 46.57 ± 0.28 | 58.10 ± 0.12 | 42.23 ± 0.28 | 51.40 ± 0.41 | 98.69 ± 0.05 | 98.54 ± 0.07 | 64.45 ± 0.71 | 69.11 ± 0.22 | 59.30 ± 0.53 | 63.88 ± 0.67 |
| | (R4Q2,R2Q1)ETU | 12.32 ± 0.46 | 17.73 ± 0.22 | 15.67 ± 0.51 | 18.42 ± 0.12 | 16.18 ± 0.25 | 25.69 ± 0.33 | 22.90 ± 0.07 | 28.57 ± 0.23 | 87.06 ± 0.28 | 89.33 ± 0.09 | 28.23 ± 0.46 | 32.38 ± 0.27 | 23.55 ± 0.35 | 28.21 ± 0.21 |
| | (R4Q2,R2Q1)VLATTACK | 36.42 ± 0.33 | 45.86 ± 0.18 | 29.13 ± 0.25 | 44.62 ± 0.11 | 40.02 ± 0.23 | 54.12 ± 0.19 | 35.60 ± 0.12 | 47.09 ± 0.27 | 94.12 ± 0.08 | 96.80 ± 0.06 | 41.81 ± 0.54 | 52.69 ± 0.29 | 38.07 ± 0.37 | 46.24 ± 0.41 |
| | (R4Q2,R2Q1)VLPTransferAttack | 38.93 ± 0.18 | 48.90 ± 0.12 | 30.50 ± 0.32 | 45.12 ± 0.11 | 48.11 ± 0.09 | 58.64 ± 0.32 | 42.53 ± 0.27 | 52.37 ± 0.76 | 99.74 ± 0.09 | **99.72 ± 0.12** | 74.24 ± 0.36 | 74.48 ± 0.17 | 70.35 ± 0.24 | 59.78 ± 0.45 |
| | (R4Q2,R2Q1)MAA | **39.70 ± 0.58** | **50.09 ± 0.35** | **32.52 ± 0.14** | **47.50 ± 0.24** | **51.02 ± 0.08** | **61.07 ± 0.11** | **44.34 ± 0.67** | **55.23 ± 0.33** | **99.78 ± 0.22** | 99.65 ± 0.30 | **76.02 ± 0.41** | **76.11 ± 0.23** | **71.19 ± 0.18** | **72.14 ± 0.17** |
| TCL | (R4Q2,R2Q1)PGD | 9.26 ± 0.25 | 13.82 ± 0.29 | 9.35 ± 0.47 | 16.52 ± 0.12 | 23.26 ± 0.24 | 38.42 ± 0.45 | 16.26 ± 0.19 | 24.51 ± 0.09 | 18.09 ± 0.16 | 24.49 ± 0.67 | 94.01 ± 0.17 | 96.83 ± 0.09 | 14.46 ± 0.37 | 18.88 ± 0.21 |
| | (R4Q2,R2Q1)BERT-Attack | 29.94 ± 0.00 | 39.21 ± 0.00 | 25.64 ± 0.00 | 39.56 ± 0.00 | 41.62 ± 0.00 | 59.57 ± 0.00 | 36.14 ± 0.00 | 48.78 ± 0.00 | 9.91 ± 0.00 | 23.64 ± 0.00 | 37.09 ± 0.00 | 53.07 ± 0.00 | 9.53 ± 0.00 | 22.80 ± 0.00 |
| | (R4Q2,R2Q1)Co-Attack | 32.20 ± 0.22 | 41.78 ± 0.17 | 27.28 ± 0.09 | 42.20 ± 0.22 | 45.89 ± 0.37 | 61.21 ± 0.30 | 40.33 ± 0.27 | 50.54 ± 0.14 | 45.09 ± 0.19 | 58.12 ± 0.38 | 94.17 ± 0.15 | 96.80 ± 0.04 | 40.30 ± 0.47 | 52.67 ± 0.84 |
| | (R4Q2,R2Q1)SGA | 39.32 ± 0.24 | 47.66 ± 0.16 | 29.92 ± 0.02 | 44.79 ± 0.13 | 48.25 ± 0.09 | 60.88 ± 0.25 | 42.62 ± 0.19 | 52.02 ± 0.31 | 68.90 ± 0.27 | 74.51 ± 0.11 | 99.15 ± 0.05 | 99.96 ± 0.02 | 63.46 ± 0.35 | 69.03 ± 0.41 |
| | (R4Q2,R2Q1)ETU | 11.59 ± 0.55 | 16.27 ± 0.32 | 24.45 ± 0.27 | 35.89 ± 0.22 | 20.06 ± 0.08 | 27.54 ± 0.12 | 19.78 ± 0.26 | 27.46 ± 0.11 | 20.18 ± 0.17 | 23.65 ± 0.22 | 91.06 ± 0.07 | 89.40 ± 0.05 | 15.49 ± 0.26 | 16.35 ± 0.47 |
| | (R4Q2,R2Q1)VLATTACK | 36.52 ± 0.67 | 21.22 ± 0.23 | 31.47 ± 0.20 | 45.75 ± 0.19 | 43.88 ± 0.11 | 54.92 ± 0.04 | 36.83 ± 0.17 | 47.78 ± 0.26 | 34.26 ± 0.13 | 46.68 ± 0.34 | 94.17 ± 0.06 | 96.81 ± 0.12 | 29.43 ± 0.42 | 42.74 ± 0.11 |
| | (R4Q2,R2Q1)VLPTransferAttack | 41.23 ± 0.34 | 48.88 ± 0.27 | 32.39 ± 0.29 | 47.98 ± 0.18 | 52.49 ± 0.06 | 62.95 ± 0.09 | 47.13 ± 0.12 | 57.14 ± 0.24 | 71.66 ± 0.06 | 75.29 ± 0.10 | 99.89 ± 0.02 | 99.70 ± 0.11 | 66.74 ± 0.27 | 70.33 ± 0.21 |
| | (R4Q2,R2Q1)MAA | **42.01 ± 0.08** | **50.98 ± 0.31** | **34.36 ± 0.44** | **50.00 ± 0.25** | **55.36 ± 0.28** | **64.27 ± 0.33** | **49.65 ± 0.58** | **57.85 ± 0.33** | **72.85 ± 0.21** | **75.69 ± 0.18** | **99.89 ± 0.11** | **99.95 ± 0.05** | **67.78 ± 0.52** | **70.75 ± 0.35** |

