# OpenReview forum: "MAA: Meticulous Adversarial Attack against Vision-Language Pre-trained Models"
_ICLR.cc/2025/Conference — Submitted to ICLR 2025_

### Official Review · Reviewer_cGtF · 2024-10-31

**Soundness:** 3
**Presentation:** 3
**Contribution:** 3
**Rating:** 5
**Confidence:** 2

**Summary:**

This paper proposes a Meticulous Adversarial Attack (MAA) method for vision-language pre-trained (VLP) models. It aims to address the limited transferability of existing adversarial attacks. MAA consists of a resizing and sliding crop (RScrop) technique and a multi-granularity similarity disruption (MGSD) strategy. The method is evaluated on various VLP models, datasets, and tasks, showing enhanced effectiveness and transferability of adversarial attacks.

**Strengths:**

- **Originality**: The proposed MAA method is novel in its combination of RScrop and MGSD strategies. The idea of exploiting model-independent characteristics and vulnerabilities of images for generating adversarial examples is original.
- **Quality**: The experiments are comprehensive, covering multiple VLP models, datasets, and downstream tasks. The ablation study and parameter analysis provide in-depth understanding of the method's components and performance.
- **Clarity**: The paper is well-written and organized. The methodology is clearly explained, and the figures and tables help in understanding the concepts.
- **Significance**: The work is significant as it addresses an important issue in the field of adversarial attacks on VLP models. The enhanced transferability of attacks can have implications for evaluating the robustness of these models.

**Weaknesses:**

- The comparison with existing methods could be more detailed. For example, a more in-depth analysis of why MAA outperforms other methods in terms of transferability could be provided.
- The text perturbations are not as thoroughly explored as the image perturbations. Since text perturbations are also an important part of multi-modal attacks, more attention could be given to this aspect.

**Questions:**

- How does the proposed method compare with other recent methods that have not been included in the comparison?
- Can the method be extended to other multi-modal tasks or domains?
- What are the potential limitations of the RScrop technique in handling very large images or complex image structures?

---

> ### Author Response · Authors · 2024-11-25
> **Response to Reviewer cGtF (1/4)**
>
> We would like to thank the reviewer for the thoughtful feedback and valuable suggestions. We have carefully addressed the comments and made revisions to improve the paper. Below, we provide our responses to the reviewer’s questions and the corresponding changes in the manuscript. We appreciate the opportunity to clarify and improve our work.
>
> **Q1**: How does the proposed method compare with other recent methods that have not been included in the comparison?
>
> **RQ1**: We have included the **most concurrent work [1]** in our revised paper and give an introduction and comparison with our method in **Table 2-5**. We also compare it with the proposed method across **different perturbation magnitudes** to further verity the effectiveness of the proposed method in Figure 2. Moreover, we have expanded our evaluation to include comparisons with methods employing **different augmentations**, such as DIM [2], TI-DIM [3], SI-NI-TI-DIM [4], and SIA [5], as shown in **Tables 6 and 10**. These additional comparisons clearly demonstrate the effectiveness of our method. The results indicate that our approach achieves state-of-the-art performance.
>
> For your reference, we have included partial results on Flickr 30K in image-text retrieval using CLIP_ViT-B/16 as the source model to illustrate the superiority of our method. **Further results and detailed analyses have been incorporated into the revised paper, highlighted in blue and labeled as R5Q1.**
>
> Table 1. Performance comparison with the most recent state-of-the-art method.
> | Target Model| Method   | Image-to-Text, R@1|  Image-to-Text, R@5|  Image-to-Text, R@10 | Text -to- Image, R@1| Text -to- Image, R@5 | Text -to- Image, R@10 |
> |:---------:|:----------:|:------:|:-----:|:------:|------:|:-----:|:------:|
> | CLIP_ViT-L/14|VLPTransferAttack |42.54 | 23.99| 14.55| 53.82 |  36.24| 28.39|
> | CLIP_ViT-L/14| MAA |54.52| 33.36| 25.76|63.80| 46.32|   38.11|
> | CLIP_RN101|  VLPTransferAttack | 51.64 | 28.45 | 19.74 | 59.42|   40.83| 33.21 |
> | CLIP_ RN101|  MAA | 72.23 | 55.34| 44.89 | 74.87|   60.65| 53.42|
> | ALBEF | VLPTransferAttack | 30.28| 12.96| 6.14| 42.88|   25.45| 19.17|
> | ALBEF|  MAA | 32.45| 14.68| 11.20| 43.62|   25.63| 19.83|
> | TCL | VLPTransferAttack | 30.66| 12.79| 6.10| 42.79|   26.52| 19.20|
> | TCL | MAA |  41.42| 22.42| 17.03  |   50.29| 31.24 |     24.82|

---

> ### Author Response · Authors · 2024-11-25
> **Response to Reviewer cGtF (2/4)**
>
> **Q2**: Can the method be extended to other multi-modal tasks or domains?
>
> **RQ2**: Yes, the proposed method is designed to be **task- and architecture-independent**, enabling transferable attacks across diverse models and tasks. To demonstrate its versatility, we have included experiments on attacking open-source large vision-language models such as **MiniGPT-4** [6] and **Llama 3.2** [7], applied to **image captioning** and **visual question answering(VQA).**
> MiniGPT-4, built on Vicuna V0 13B (a 13-billion-parameter large language model), has scaled up the capabilities of large language models and shows performance comparable to GPT-4. For Llama 3.2, we utilized Llama-3.2-11B-Vision-Instruct, which comprises 11 billion parameters.
> Specifically, MiniGPT-4, based on Vicuna V0 13B (a 13-billion-parameter large language model), has recently scaled up the capabilities of large language models and demonstrates performance comparable to GPT-4. For Llama 3.2, we utilized Llama-3.2-11B-Vision-Instruct, which comprises 11 billion parameters.
> In the image captioning task, we provided images with the prompt: “Describe this image, bringing all the details” as input. For the VQA task, we paired images with the question: “What is the content of this image?” and collected the generated descriptions and answers for each attack method.
>
> To assess attack performance, we used the CLIP score [8], which measures the similarity between the features of the generated descriptions/answers for adversarial images and their clean counterparts, as processed by the CLIP text encoder. To ensure fair comparisons, we calculated the CLIP score between the adversarial image features (generated by each attack method) and the clean image features, as large vision-language models may produce different responses to the same input at different query times. Additionally, we reported a baseline CLIP score by comparing the features of clean images with those of their randomly shuffled counterparts, providing a reference point to highlight the effectiveness of the attacks.
>
> The results, provided below, demonstrate that our proposed method achieves the best performance across both tasks, further validating its transferability. However, we note that attack performance is relatively modest due to the substantial differences between the source and target models in terms of architecture, training data, and methodologies. Improving attack performance on large-scale models remains an open challenge, which we identify as a promising direction for future research.
> **Detailed experimental information, including settings, results, and corresponding analysis, has been added to Section A.3 of the revised paper. These additions are highlighted in blue and labelled as R5Q2.**
>
> Table 2. Evaluating robustness of Llama 3.2 in vision question answering, different CLIP text encoders are used to generate text features for similarity measurement
>
> | Method    | CLIP_ViT-B/16 | CLIP_ViT-B/32 | CLIP_ ViT-L/14 | CLIP_ RN50 | CLIP_ RN101| Average|
> |:---------:|:-----:|:------:|:-----:|:------:|:-----:|:------:|
> | Random Shuffle | 0.415| 0.417 | 0.312 | 0.380|   0.507| 0.406|
> | Co-Attack| 0.817 | 0.814  | 0.784| 0.802|   0.840 | 0.811  |
> | SGA | 0.815 | 0.811 | 0.783|   0.801| 0.839|   0.810|
> | ETU| 0.823  | 0.820| 0.792|   0.810| 0.845| 0.818|
> | VLATTACK| 0.812 | 0.809| 0.781| 0.799|   0.838| 0.808 |
> | VLPTransferAttack | 0.810  | 0.806| 0. 776| 0. 794|   0. 835| 0. 804|
> | MAA| 0. 798| 0. 794| 0. 763| 0. 782|   0. 825| 0. 793|
>
> Table 3. Evaluating robustness of MiniGPT-4 in image captioning, different CLIP text encoders are used to generate text features for similarity measurement
>
> | Method    | CLIP_ViT-B/16 | CLIP_ViT-B/32 | CLIP_ ViT-L/14 | CLIP_ RN50 | CLIP_ RN101.| Average|
> |:---------:|:-----:|:------:|:-----:|:------:|:-----:|:------:|
> | Random Shuffle | 0. 445| 0. 427| 0. 302| 0. 398|   0. 631| 0. 441|
> | Co-Attack | 0. 856| 0. 823| 0. 771| 0. 847|   0. 875| 0. 834|
> | SGA | 0. 862| 0. 818| 0. 775|   0. 842| 0. 869|   0. 834|
> | ETU| 0. 880| 0. 825| 0. 792|   0. 862| 0. 883| 0. 848|
> | VLATTACK| 0. 853| 0. 815| 0. 780| 0. 843|   0. 874| 0. 833|
> | VLPTransferAttack | 0. 852| 0. 811| 0. 776| 0. 842|   0. 870| 0. 830|
> | MAA| 0. 843| 0. 801| 0. 768| 0. 833|   0. 859| 0. 821|

---

> ### Author Response · Authors · 2024-11-25
> **Response to Reviewer cGtF (3/4)**
>
> **Q3**: What are the potential limitations of the RScrop technique in handling very large images or complex image structures?
>
> **RQ3**: Thank you for highlighting this important consideration. The proposed method is designed to address complex black-box application scenarios, where target models and tasks are unknown. Different tasks often require models to prioritize varying aspects of the data. For instance, in image retrieval, models focus on matching entire images to text, whereas tasks such as visual grounding and image captioning emphasize fine-grained details, such as objects or locations relevant to textual descriptions. This necessitates that attackers account for both global information and fine-grained details within the visual input.
>
> Handling very large images or complex image structures poses additional challenges, as it requires deeper exploration of the information embedded in the data. Current methods often overlook fine-grained image details, limiting their effectiveness in such scenarios. To address this, our proposed method incorporates the RScrop technique, which performs resizing and sliding cropping on images. Coupled with the multi-granularity similarity disruption strategy, RScrop enables the investigation of detailed information, showing potential for managing very large images and complex image structures.
>
> One potential improvement for the proposed method would be that very large images or complex image structures may require lager scaling factors to focus more effectively on the fine-grained details and better exploit the characteristics and vulnerabilities of such images. We recognize this as an exciting direction for future research and appreciate the opportunity to discuss it further.
>
>
> [1] Gao, Sensen, et al. "Boosting transferability in vision-language attacks via diversification along the intersection region of adversarial trajectory." ECCV. 2024: 442-460.
>
> [2] Xie, Cihang, et al. "Improving transferability of adversarial examples with input diversity." CVPR2019.
>
> [3] Dong, Yinpeng, et al. "Evading defenses to transferable adversarial examples by translation-invariant attacks." CVPR 2019
>
> [4] Lin, Jiadong, et al. "Nesterov accelerated gradient and scale invariance for adversarial attacks." ICLR2020
>
> [5] Wang, Xiaosen, Zeliang Zhang, and Jianping Zhang. "Structure invariant transformation for better adversarial transferability." CVPR2023
>
> [6] Deyao Zhu, Jun Chen, Xiaoqing Shen, Xiang Li, and Mohamed Elhoseiny. Minigpt-4: Enhancing vision-language understanding with advanced large language models. arXiv preprint arXiv:2304.10592, 2023.
>
> [7] Touvron, Hugo, et al. "Llama: Open and efficient foundation language models." arXiv preprint arXiv:2302.13971 (2023).
>
> [8] Zhao, Yunqing, Tianyu Pang, Chao Du, Xiao Yang, Chongxuan Li, Ngai-Man Cheung, and Min Lin. "On evaluating adversarial robustness of large vision-language models." In Proceedings of the 37th International Conference on Neural Information Processing Systems, pp. 54111-54138. 2023.

---

> ### Author Response · Authors · 2024-11-25
> **Response to Reviewer cGtF (4/4)**
>
> **W1** : The comparison with existing methods could be more detailed. For example, a more in-depth analysis of why MAA outperforms other methods in terms of transferability could be provided.
>
> **RW1** : Thank you for this valuable suggestion. The superior performance of our proposed method is partially rooted in a well-established insight: tailored perturbations for individual images tend to be more effective than applying uniform perturbations across all images [1,2].
>
> Different tasks require models to focus on varying aspects of the data. For instance, in image retrieval, models emphasize matching entire images to text, while tasks such as visual grounding and image captioning focus on fine-grained details like objects or locations within the visual input that correspond to textual descriptions. This necessitates attackers to account for both global information and fine-grained details simultaneously.
>
> To address this, we introduced a novel combination of resizing and sliding crop (RScrop) and multi-granularity similarity disruption (MGSD) strategies, which complement each other. MGSD facilitates the exploration of local regions by extracting features from diverse intermediate layers, while RScrop mitigates the limitations of fixed-size local region processing in standard deep models. By capturing more detailed information and local dependencies, RScrop ensures comprehensive image coverage across layers. Together, these techniques enable a deeper exploration of fine-grained characteristics and vulnerabilities of the input samples.
> In contrast, existing methods primarily rely on data augmentation to increase data diversity and reduce overfitting. However, these approaches do not effectively explore the fine-grained details of samples, which limits their performance and results in lower transferability.
>
> Additionally, we have conducted further comparisons with popular existing augmentation methods to highlight the effectiveness of our approach. Detailed analyses and experimental results have been included in the revised paper to enhance clarity and understanding. These updates are highlighted in blue and labelled as **R5W1**.
>
> [1] Omid Poursaeed, Isay Katsman, Bicheng Gao, and Serge Belongie. Generative adversarial perturbations. In Proceedings of the IEEE Conference on Computer Vision and Pattern Recognition, pp. 4422–4431, 2018.
>
> [2] Muzammal Naseer, Salman Khan, Muhammad Haris Khan, Fahad Shahbaz Khan, and Fatih Porikli. Cross-domain transferability of adversarial perturbations. In Proceedings of the International Conference on Neural Information Processing Systems, pp. 12905–12915, 2019.
>
> **W2** : The text perturbations are not as thoroughly explored as the image perturbations. Since text perturbations are also an important part of multi-modal attacks, more attention could be given to this aspect.
>
> **RW2** : Thank you for this thoughtful suggestion. In this paper, our primary focus is on improving attack effectiveness by enhancing image perturbations, based on the observation that current methods often rely heavily on text perturbations to achieve effective attacks. For the text modality, we employ the widely used BERT-Attack method. To address your feedback, we have added more details about this approach in the revised paper, specifically **under Eq. 4**, highlighted in blue and labelled as **R5W2**. The added explanation is as follows:
>
>  “BERT-Attack first identifies the most important word in each sentence by replacing each word in the sentence with [MASK] one at a time and ranking the feature distance between each modified sentence with the original sentence and the paired image. The most important word would be replaced by a semantic-consistent word to ensure visually plausible attacks. For the attack effectiveness, BERT is used to generate a set of candidates and the one that can minimize the loss in Eq.4 would be selected to replace the original word to realize an attack”.

---

> ### Author Response · Authors · 2024-11-28
> **A Kind Reminder: Follow-Up on Revisions and Responses**
>
> Dear Reviewer cGtF,
>
> Thank you once again for your time and insightful suggestions on our paper!
>
> As the discussion period comes to an end, we would like to check whether our responses and the revised paper have adequately addressed your concerns. Your feedback is invaluable to us, and we would greatly appreciate the opportunity to engage in further discussions if needed.
>
> Sincerely, The Authors

---

> > ### Author Response · Authors · 2024-12-03
> > **A Kind Reminder: Discussion Period Comes to An End**
> >
> > Dear Reviewer cGtF,
> >
> > Thank you once again for your valuable guidance in enhancing the quality and scope of our paper.
> >
> > As the discussion period comes to an end, we would like to confirm whether our response has adequately addressed your remaining concerns. If you are satisfied with our clarifications, we would be truly grateful if you could kindly raise your score.
> >
> > Best regards,
> > The Authors

---

### Official Review · Reviewer_3fPG · 2024-10-31

**Soundness:** 2
**Presentation:** 2
**Contribution:** 3
**Rating:** 5
**Confidence:** 5

**Summary:**

In this paper, the authors propose a new attack method called the Meticulous Adversarial Attack (MAA) that focuses on exploiting model-independent characteristics and vulnerabilities of images. The approach incorporates a novel resizing and sliding crop (RScrop) technique along with a multi-granularity similarity disruption (MGSD) strategy, aiming to overcome the limitations of existing VLP models in capturing detailed aspects and their connections. The experimental results have demonstrated the effectiveness of the proposed method.

**Strengths:**

1.	The experimental setup in this paper is comprehensive, demonstrating the effectiveness of the proposed method.
2.	This paper is easy to follow.

**Weaknesses:**

1.	The data augmentation method based on scaling and the joint attack on shallow features proposed in this paper for the image modality are indeed scenario techniques in the field of adversarial examples. Similarly, the method for the text modality is an existing technique.
2.	Although the authors have provided the motivation behind their proposed scheme, the absence of corresponding formula analysis and illustrative figures makes it rather difficult to understand. Moreover, the authors lack in-depth insights to analyze the effectiveness of the proposed RScrop and MGSD strategies.
3.	The main attack algorithm logic in the code provided by the authors is chaotic and cannot be reproduced. Moreover, it fails to correspond to the method section in the paper. The overall code seems to be adapted from the SGA repository. This makes me doubt the effectiveness of the method proposed in this paper.
4.	The experiments in this paper do not provide error bars or results from experiments with different random seeds, which raises my concerns about the validity and stability of the experimental outcomes.
5.	There is a symbolic expression error in Formula 1. It should be $f_{img}^{i}(x_{k})$ instead of $f_{img}^{i}(t)$.

**Questions:**

See Weakness

---

> ### Author Response · Authors · 2024-11-25
> **Response to Reviewer 3fPG (1/3)**
>
> We would like to thank the reviewer for the thoughtful suggestion on our paper, especially for the presentation and experiments. We have carefully addressed the concerns and made revisions to improve the paper. Below are our response to the questions and weakness. We welcome any further suggestions or questions you may have and look forward to addressing them.
>
> **W1**: The data augmentation method based on scaling and the joint attack on shallow features proposed in this paper for the image modality are indeed scenario techniques in the field of adversarial examples. Similarly, the method for the text modality is an existing technique.
>
> **RW1**: This work addresses a critical limitation in existing attacks against VLP models, particularly the weakness of adversarial attacks in the image modality, which fails to effectively disrupt the interactions between images and texts. Enhancing adversarial images is crucial, as the image modality plays a central role in current multi-modal applications. This includes tasks that use both images and text as inputs, such as image-text retrieval, as well as tasks that rely solely on images, like image captioning.
>
> Different tasks require different types of information processing, making transferrable adversarial attacks across different models and tasks particularly challenging. For example, in image retrieval, models match entire images to text, while in visual grounding and image captioning, models focus on fine-grained details such as objects or locations within the image relevant to textual descriptions. This necessitates that attackers consider both global information and finer details. Moreover, in multi-modal scenarios, the cross-modal interactions should also be addressed. In our experiments, we show that compared methods are less effective to deal with this.
>
> Considering this, we propose leveraging the RScrop augmentation and the multi-granularity similarity disruption strategy to break the intra- and inter-model relationships across various granularities and hierarchical levels. By resizing images and applying sliding crops, allowing the model to focus on detailed regions and their dependencies while ensuring comprehensive image coverage across layers. This process captures finer-grained characteristics and vulnerabilities of the input, and also create more diverse data to explore cross-modal interactions, thereby enhancing attack performance across different models and vision-language tasks. Our experiments show that the proposed method significantly disrupt cross-modal interactions across a range of models and tasks, demonstrating its effectiveness in adversarial attacks against VLP models.
>
> While text attacks are equally important for many scenarios, we plan to explore them in future work.

---

> ### Author Response · Authors · 2024-11-25
> **Response to Reviewer 3fPG (2/3)**
>
> **W2**: Although the authors have provided the motivation behind their proposed scheme, the absence of corresponding formula analysis and illustrative figures makes it rather difficult to understand. Moreover, the authors lack in-depth insights to analyze the effectiveness of the proposed RScrop and MGSD strategies.
>
> **RW2**:  Thank you for the suggestion.
>
> (1) To offer a more detailed explanation of RScrop and MGSD, we have expanded **Section 2.2**, highlighted in blue and annotated as **R4W2**. Specifically, we made following modifications on the paper:
>
> - We add a clear formulation to define how to set step size of each sliding crop operation: "The shift step length for i-th step relative to the initial point in different dimensions can be formulated by $L_{x/y}^i= (i/2) * l_{x/y} + (i$\%$2) *α_{x/y}^i, α_{x/y}^i = UniformDiscrete(β_1, β_2)$, where $x$ and $y$ denote $x$- and $y$- dimension. $l_{x/y}$ is the shift window length for x- or y-dimension.  $β_1, β_2$ are smaller than the size of dimension size of the patch or the convolutional filter. "
>
> - We also add content how to construct a set of adversarial examples through RScrop for optimizing final adversarial images "Formally, after Rscrop, we can obtain a set of transformed adversarial images $\mathcal{R}({x}^{\text{adv}}) =$ { ${{x}_1^{\text{adv}}, {x}_2^{\text{adv}}, ..., {x}_k^{\text{adv}}}$ }. We maximize the feature distance between all these adversarial images and original images at various layers and components as follows".
>
> - Under Eq. 1, we explain the feature extraction process from different kinds of CLIP models: "For vision transformer-based VLP models, features would include those output from each self-attention module and final output layer, while for ResNet-based VLP models, we extract features from each residual block and final output layer. By focusing on features across different layers and different scales of images, we are able to explore fine-grained vulnerabilities of samples."
>
> - We provide a detailed explanation of BERT-Attack for adversarial text generation. For example, under Formula (3), we provide more details about the BERT-Attack used for generating adversarial texts: "BERT-Attack first identifies the most important word in each sentence by replacing each word with [MASK] one at a time and ranking the feature distance between each modified sentence and the original sentence paired with the image. The most important word is then replaced by a semantic-consistent word to ensure visually plausible attacks. For attack effectiveness, BERT generates a set of candidate words, and the one that minimizes the loss in Eq. (4) is selected to replace the original word to realize the attack."
>
> - We further explain how the proposed method enlarge the gap between image and texts to promote effective attacks, e.g., "The RScrop also creates diverse image-text pairs, which help better explore cross-model interactions for transferrable attacks. We enlarge the feature distance between adversarial images and their original paired texts to comprehensively disturb image-text connections."
>
> (2) Furthermore, in Section 3.4, we conduct **ablation studies** to assess the contribution of RScrop and MGSD, demonstrating that these two methods complement each other to achieve superior performance. To analyze their effectiveness, we compare RScrop with various other augmentations, including DIM [1], TI-DIM [2], SI-NI-TI-DIM [3], and SIA [4], with results presented in **Tables 6 and 10**. From the results, it is clear that RScrop outperforms other augmentations and complements MGSD, helping to capture more fine-grained information.
>
> (3) In addition, we present **experiments and visualizations** that test the performance of ALBEF in visual grounding with and without the proposed augmentation, as shown in **Section A.2** of the revised paper. These results demonstrate that the proposed method can extract more overlapping regions with the ground-truth, highlighting the capture of contextual features. All of these findings provide strong evidence for the effectiveness of the proposed method.
>
> [1] Xie, Cihang, et al. "Improving transferability of adversarial examples with input diversity." CVPR2019.
>
> [2] Dong, Yinpeng, et al. "Evading defenses to transferable adversarial examples by translation-invariant attacks." CVPR 2019
>
> [3] Lin, Jiadong, et al. "Nesterov accelerated gradient and scale invariance for adversarial attacks." ICLR2020
>
> [4] Wang, Xiaosen, Zeliang Zhang, and Jianping Zhang. "Structure invariant transformation for better adversarial transferability." CVPR2023

---

> ### Author Response · Authors · 2024-11-25
> **Response to Reviewer 3fPG (3/3)**
>
> **Q1**: The main attack algorithm logic in the code provided by the authors is chaotic and cannot be reproduced. Moreover, it fails to correspond to the method section in the paper. The overall code seems to be adapted from the SGA repository. This makes me doubt the effectiveness of the method proposed in this paper.
>
> **RQ1**: We appreciate the reviewer’s careful examination of our code and methodology. While our implementation is inspired by SGA, the core logic and components of our attack algorithm, such as RScrop and MGSD, are novel and distinct from SGA. To address the concerns regarding clarity and alignment with the paper, we have reorganized the code to include comprehensive documentation and step-by-step instructions. This will ensure that the implementation is easier to follow and corresponds precisely to the methodology described in the paper. The updated and well-organized code will be uploaded and made available shortly.
>
> **Q2**: The experiments in this paper do not provide error bars or results from experiments with different random seeds, which raises my concerns about the validity and stability of the experimental outcomes.
>
> **RQ2**: Thank you for your valuable suggestion. We have conducted additional experiments **using multiple random seeds (i.e., [42, 123, 256, 789, 1001])** to evaluate the stability of our results. Error bars have been incorporated into the updated results (Tables 2-5) to better reflect the variability and robustness of the experimental outcomes. The results show that the proposed method is stable and provides consistent improvements over the compared methods. These updates will be included in the revised version of the paper for clarity and transparency, highlighted in blue and annotated as **R4Q2.**
>
> **Q3**: There is a symbolic expression error in Formula 1. It should be fimgi(xk) instead of fimgi(t).
>
> **RQ3**: Thank you for pointing that out. We will revise the formula and thoroughly review the presentation to ensure accuracy and improve clarity for better understanding.

---

> ### Author Response · Authors · 2024-11-28
> **A Kind Reminder: Follow-Up on Revisions and Responses**
>
> Dear Reviewer 3fPG,
>
> Thank you once again for your time and insightful suggestions on our paper!
>
> As the discussion period comes to an end, we would like to check whether our responses and the revised paper have adequately addressed your concerns. Your feedback is invaluable to us, and we would greatly appreciate the opportunity to engage in further discussions if needed.
>
> Sincerely, The Authors

---

### Official Review · Reviewer_PuP3 · 2024-11-03

**Soundness:** 3
**Presentation:** 2
**Contribution:** 3
**Rating:** 8
**Confidence:** 4

**Summary:**

This paper proposes a novel transferable adversarial attack method for VL models called Meticulous Adversarial Attack (MAA).

The idea is to minimize dependency on the surrogate model and exploit model-independent features using image augmentations.
This paper proposes a simple resizing and sliding crop (RScrop) technique for image augmentation.
MAA also applies multi-granularity similarity disruption (MGSD), which is similar to the existing VLAttack.

MAA achieves SOTA attack success rate in Image-Text Retrieval, Visual Grounding, and Image Captioning tasks.

**Strengths:**

- The transferability improvement compared to the SOTA attacks is significant.
- The method is simple, combining existing ideas to improve the transferability: utilizing image augmentations and multi-granularity similarity disruption (MGSD).

**Weaknesses:**

- The source model is fixed to CLIP ViT-B/16. Following existing work, it is better to provide results for the other source models for completeness.

- I expect a deeper analysis of the effectiveness of the proposed RScrop. Table 6 should be improved. I am unsure of the contribution and the novelty of RScrop without comparisons between different image augmentation techniques.
    - While the design of the proposed RScrop (resize and slide) seems reasonable, how is it better than other input transformations used in existing methods? What about a comparison between “random resize and padding” (DIM [A]), “translation” (TI-DIM [B]), and “scale and translation” (SI-NI-TI-DIM [C])? Also, the latest method, SIA [D], introduces many different types of augmentations, such as Shift, Rotate, Scale, Noise, etc.
    - The claim that the RScrop is effective because it enables the model “to attend to intricate local details” seems unverified. Since RScrop only zooms in (scale factor > 1.0), a comparison between Unzooming (scale factor < 1.0) / Random scale (e.g., scale factor ranging from 0.5 ~ 2.0) can verify this.
    - “To attend to previously overlooked boundary regions of adjacent patches,” existing augmentations, such as random resize and padding, or translation can be a solution. What are the differences between these augmentations and RScrop?
    - The connection between image augmentation techniques and MGSD is unclear. Does the effectiveness of each image augmentation technique differ when MGSD is applied or not?
- minor
    - A typo in Equation (1): $f^i_{img}$ should input $x$.
    - The citation style is wrong: Authors should correctly use \citet and \citep.

---

[A] Xie, Cihang, et al. "Improving transferability of adversarial examples with input diversity." *CVPR*2019.

[B] Dong, Yinpeng, et al. "Evading defenses to transferable adversarial examples by translation-invariant attacks." *CVPR 2019*

[C] Lin, Jiadong, et al. "Nesterov accelerated gradient and scale invariance for adversarial attacks." ICLR2020

[D] Wang, Xiaosen, Zeliang Zhang, and Jianping Zhang. "Structure invariant transformation for better adversarial transferability." *CVPR2023*

**Questions:**

- What is the computational cost of MAA? It uses K augmentations and many iterations (60), so I assumed it was large.

---

> ### Author Response · Authors · 2024-11-25
> **Response to Reviewer PuP3 (1/3)**
>
> We would like to thank the reviewer for the thoughtful suggestion on our paper, especially for verifying the effectiveness of each component and their complementary nature. We have carefully addressed the concerns and made revisions to improve the paper. Below are our response to the questions and weakness. We welcome any further suggestions or questions you may have and look forward to addressing them.
>
> **W1**: The source model is fixed to CLIP ViT-B/16. Following existing work, it is better to provide results for the other source models for completeness.
>
> **RW1**: Thank you for the suggestion. In Table 2 of the paper, we evaluate adversarial examples generated using CLIP ViT-B/16 as the source model. To provide a more comprehensive analysis, **we include results from additional source models, such as CLIP ResNet101, ALBEF, and TCL, in Table 3.**
>
> Moreover, for tasks like visual grounding and image captioning, we have incorporated results using ALBEF as the source model. These results consistently demonstrate the effectiveness of the proposed methods across various source models. **The added content is highlighted in blue and annotated as R3W1 in the revised paper.**
>
> **W2.1**: comparison with other augmentation. While the design of the proposed RScrop (resize and slide) seems reasonable, how is it better than other input transformations used in existing methods? What about a comparison between “random resize and padding” (DIM [A]), “translation” (TI-DIM [B]), and “scale and translation” (SI-NI-TI-DIM [C])? Also, the latest method, SIA [D], introduces many different types of augmentations, such as Shift, Rotate, Scale, Noise, etc.
>
> **RW2.1**: Thank you for the thoughtful suggestions. We have conducted experiments incorporating all the suggested augmentations and added the results to **Table 6 and Table 10 in Section A.4.** These updates are highlighted in blue and annotated as R3W2.1 in the revised paper. For your convenience, we also **present the results below for reference.**
>
> Table 1. Comparison with different augmentations.
> | Target Model   | CLIP_ViT-B/16. | CLIP_ ViT-L/14| CLIP_ RN101 | ALBEF | TCL |
> |:---------:|:-----:|:------:|:-----:|:------:|:------:|
> | Source Model    |I2T, T2I| I2T, T2I | I2T, T2I. | I2T, T2I  | I2T,T2I  | I2T, T2I  |
> | MAA w DIM | 99.63, 99.87 | 42.12, 53.09 |   62.45, 65.13 |   27.22, 41.3 |   31.61, 44.5 |
> | MAA w TI-DIM |  96.13,  99.27|  32.02,  46.3|   41.89, 53.14| 24.81, 31.62  |   24.75,  29.6|
> | MAA w SI-NI-TI-DIM | 99.85,  99.19| 26.99,  41.59  |   34.11,  4.91|  28.95,  33.51| 22.64 , 29.98|
> | MAA w SIA | 99.51,  99.23| 42.94,  54.25|   63.22, 69.19|    27.42, 40.79|    35.09,  45.31|
> | MAA w DIM  w/o MGSD| 100.00,  100.00| 41.37, 54.72|   62.96,  63.48|  26.13, 40.5|   29.98,  42.30|
> | MAA w TI-DIM w/o MGSD |  99.14, 99.36| 38.53,  52.35|   44.83, 56.4|   25.03, 30.64| 23.12,  27.23
> | MAA w SI-NI-TI-DIM w/o MGSD | 99.94,  99.57| 27.48, 42.11 |   34.87,  44.01|  27.48,  33.97| 22.14 ,   28.42|
> | MAA w SIA w/o MGSD | 98.89,  99.22| 41.11,  53.56|   61.66, 68.81|     25.75,  40.74|  33.24, 42.80|
> | MAA w/o RScrop | 100.0,  100.00| 25.47, 41.85|   38.83,  49.23|  12.62 ,  27.48| 13.9,  28.9|
> | MAA w RScrop w/o MGSD | 99.51,  99.71| 45.02,  56.55|   57.81,  65.82|    27.63,  43.59|    32.77 ,   45.48|
> | MAA| 100.00,  99.97| 54.36, 63.02|   73.05,  74.85|    31.78,  43.62|    41.07,  49.81  |
>
> To provide a deeper understanding of this line of work and demonstrate the superiority of the proposed method, we have also included a detailed analysis in **Section A.4**:
>
> “From the results, several key observations emerge. First, data augmentations consistently enhance attack performance compared to scenarios where they are not utilized. This improvement is due to the increased data diversity, which prevents overfitting to source models. For instance, transformations such as translation or rotation modify the geometric properties of an image, ensuring that models do not rely solely on specific discriminative regions for predictions. Second, the proposed method outperforms the compared techniques. This advantage arises because the resizing and sliding crop operations in the proposed RScrop not only capture more fine-grained details but also ensure comprehensive coverage of the image. Moreover, the augmentation complements the MGSD strategy effectively by relieving it from fixed-size local processing and MGSD provide more information about local regions, providing more valuable insights into fine-grained characteristics and vulnerability of samples. Together, these components enable the proposed method to achieve superior performance.”

---

> ### Author Response · Authors · 2024-11-25
> **Response to Reviewer PuP3 (2/3)**
>
> **W2.2**: comparison between Unzooming (scale factor < 1.0) / Random scale
>
> **RW2.2**:  In Section 3.3.2, we analyze the effects of different scaling factors, including ranges such as 1.25–1.5, 1.25–2, 2–5, and 5–10. These experiments demonstrate that appropriate resizing effectively captures fine-grained details without degrading data quality. In the revised paper, we have also added results for Unzooming and Random Scale (0.5–2.0), which are now included in Figure 3(a). From these additional results, we observe the following. First, scaling factors < 1 do not improve performance significantly, as they fail to effectively explore fine-grained details of the images. Nevertheless, these transformations still enhance performance compared to methods without augmentations. This improvement arises from the increased data diversity introduced by Unzooming and Random Scaling, which helps mitigate overfitting to source models to a certain extent. This analysis underscores the importance of proper scaling in effectively leveraging fine-grained information for improved attack performance. **The added experimental results and analysis have been added into paper, highlighted in blue and annotated as R3W2.2.**
>
> **W2.3**: “To attend to previously overlooked boundary regions of adjacent patches,” existing augmentations, such as random resize and padding, or translation can be a solution. What are the differences between these augmentations and RScrop?
>
> **RW2.3**: Random resize and padding adjust the size of images while adding zero-padding around the input, and translation shifts images within the frame. These methods help capture boundary information to some extent, addressing overlooked regions at patch edges.
>
> However, they are limited in effectively exploring fine-grained details. In contrast, the proposed RScrop utilizes resizing and sliding crop techniques to focus on smaller regions and their dependencies, enabling it to identify more nuanced image features. Additionally, by systematically sliding across the image, RScrop ensures comprehensive coverage, minimizing blind spots.
> To substantiate these claims, we compared the proposed method with these augmentations, as detailed in our response to R2.1. The consistent performance improvements demonstrate that the proposed method effectively captures more contextual information.
>
> To further validate this, we have added additional experiments and visualizations in **Section A.2** of the revised paper, showcasing RScrop’s ability to enhance contextual understanding. The observed performance gains further affirm the superiority of our method.
>
> For better reference, we present **experiments using the ALBEF model for visual grounding**, comparing its performance with and without the proposed augmentation. The evaluation metric measures the overlap between predicted and ground-truth regions, considering regions matched if the overlap exceeds a threshold (e.g., 0.5). The results are summarized below:
>
> Table 2.Performance on visual grounding.
> | | Val | TestA | TestB |
> |:----:|:-----:|:------:|:-----:|
> | Model w/o the proposed method. | 51.24 | 56.71   | 44.79 |
> | Model w/ the proposed method |  56.67 | 65.37| 45.28 |
>
> These results clearly demonstrate that the proposed method enhances overlap with ground-truth regions, highlighting its capability to extract finer-grained contextual features effectively.
>
> [A] Xie, Cihang, et al. "Improving transferability of adversarial examples with input diversity." CVPR2019.
>
> [B] Dong, Yinpeng, et al. "Evading defenses to transferable adversarial examples by translation-invariant attacks." CVPR 2019
>
> [C] Lin, Jiadong, et al. "Nesterov accelerated gradient and scale invariance for adversarial attacks." ICLR2020
>
> [D] Wang, Xiaosen, Zeliang Zhang, and Jianping Zhang. "Structure invariant transformation for better adversarial transferability." CVPR2023

---

> ### Author Response · Authors · 2024-11-25
> **Response to Reviewer PuP3 (3/3)**
>
> **W2.4**:  The connection between image augmentation techniques and MGSD is unclear. Does the effectiveness of each image augmentation technique differ when MGSD is applied or not?
>
> **RW2.4**: To clarify the relationship between various augmentations and MGSD, we conducted extensive experiments, detailed in Section A2.2 of the revised paper. Results can also be found in Table 1 under our response to W2.1.
>
> The findings reveal that MGSD complements image augmentations by leveraging features from intermediate layers, providing additional contextual information. However, the combined use of MGSD and RScrop achieves superior performance due to their complementary nature. Specifically, MGSD enhances the exploration of local region features by extracting diverse information from intermediate layers. Nonetheless, MGSD alone is constrained by the fixed-size local processing in lower layers of standard deep models, which limits its ability to capture high-level local information from the output layer, as these models process the entire image.
> RScrop addresses this limitation by resizing images and applying sliding crops, allowing the model to focus on detailed regions and their dependencies while ensuring comprehensive image coverage across layers. This process captures finer-grained characteristics and vulnerabilities of the input, thereby enhancing attack performance across different models and tasks.
> **Experimental results and detailed analyses supporting these claims are provided in Section A4of the revised paper, highlighted in blue and annotated as R3W2.1.**
>
> **Q1**: What is the computational cost of MAA? It uses K augmentations and many iterations (60), so I assumed it was large.
>
> **RQ1**. The proposed method is designed for complex black-box scenarios, where the target models and tasks are unknown. Different tasks require models to process various types of information in the data. For example, in image retrieval, models focus on matching entire images to text, while in visual grounding and image captioning tasks, models emphasize fine-grained details such as objects or locations within the image that are relevant to textual descriptions. This necessitates that attackers consider not only global information but also these finer details.
>
> To address this challenge, we propose leveraging RScrop and the multi-granularity similarity disruption (MGSD) strategy to break intra- and inter-model relationships across different granularities and hierarchical levels. While this approach requires more iterations and computational time compared to existing methods, the trade-off is justified by the substantial performance improvements it provides. For instance, on a workstation with a Tesla V100 GPU, the proposed method takes approximately 13.8 seconds to generate an adversarial image, while the SGA method takes 2.5 seconds. However, given significant performance gains, e.g., 40.6% in image-text retrieval, we believe the additional computational cost is acceptable.  In addition, SGA utilizes three-stage learning for adversarial text generations, incurring additional computational costs.

---

> ### Author Response · Authors · 2024-11-28
> **A Kind Reminder: Follow-Up on Revisions and Responses**
>
> Dear Reviewer PuP3,
>
> Thank you once again for your time and insightful suggestions on our paper!
>
> As the discussion period comes to an end, we would like to check whether our responses and the revised paper have adequately addressed your concerns. Your feedback is invaluable to us, and we would greatly appreciate the opportunity to engage in further discussions if needed.
>
> Sincerely, The Authors

---

> ### Comment · Reviewer_PuP3 · 2024-11-28
> **Response by Reviewer**
>
> Thank you for your response.
>
> [W1] Thank you, this is addressed properly.
>
> [W2-1]  Thank you for providing Tables 6 and 10. I have a few minor suggestions:
> - The term "with" should be abbreviated as "w/" instead of "w."
> - The method names are somewhat confusing. If I understand correctly, "MAA w/ DIM" means "MAA - RSCrop + DIM."
>    - For better organization, consider using two separate columns, such as MGSA and Aug.Type.
>
> [W2-2] Thank you. The claim that fine-grained details are important seems empirically supported.
>
> [W2-3] Thank you for your explanation. I now understand that "comprehensive coverage" is a key difference, and the ALBEF experiment is interesting.
>
> Based on this, "comprehensive coverage" is an important factor alongside "fine-grained details." To verify this claim, I suggest a more detailed analysis of sliding, similar to the experiment shown in Figure 3(a). For example:
> - Compare "Sliding" with "Random selection of patches (Random Sliding)" while keeping the number of patches fixed to determine whether full coverage is crucial.
> - Question: In Table 6, what does "w/o Sliding" actually mean? If it involves using only one randomly selected patch, I believe this does not verify the claim, as the number of patches differs.
>
> [W2-4] Thank you. The observation that RSCrop is particularly effective when combined with MGSD should be emphasized or explained more thoroughly. The current explanation feels lacking in theoretical depth.
>
> [Q1] Thank you. I agree that the increase in computational cost is acceptable for better evaluating the VL model's vulnerability. Nevertheless, this should be added to the Appendix.
>
> Overall,
> - I appreciate the novel augmentation design and the significant performance improvements.
> - However, since the fundamental ideas of (1) using augmentation and (2) using multiple-layer outputs are not novel, I expect more theoretical depth for the top-tier conference. The current version seems to lack,
>   - Sufficient evidence that "full coverage" is essential.
>   - Sufficient explanation/analysis of why RSCrop is particularly important for MGSD.
>
> For the current version, I believe the reasons for acceptance outweigh those for rejection. However, I will retain my score as "6: marginally above the acceptance threshold".
> I am open to reviewing additional results and engaging in further discussion.

---

### Official Review · Reviewer_y1vj · 2024-11-04

**Soundness:** 3
**Presentation:** 2
**Contribution:** 3
**Rating:** 6
**Confidence:** 3

**Summary:**

This paper presents a new approach to enhance the transferability of adversarial examples across pre-trained vision-language models. The method improves the robustness of a generated adversarial example by (1) forcing the surrogate model to attend to intricate local details and previously overlooked boundary regions, and (2) optimizing feature distances at both low-level (local regions) and high-level (semantic). Evaluation shows that the proposed method can significantly outperform state-of-the-art methods across a large set of vision-language models. Ablation study shows that both techniques contributes to the improved performance.

**Strengths:**

+ The proposed techniques make sense
+ The evaluation is thorough
+ The evaluation results are good

**Weaknesses:**

- The proposed method requires more computation power (iterations), which would either increase the delay of the attack (making it less ideal for real-time attacks), or require more powerful computers
- Performance on BLIP seems not as good
- The writing could be improved. For example, I don't think formula (1) matches its description.

**Questions:**

* Please explain why BLIP is excluded from I2T and T2I retrieval experiments, why visual grounding is only done with ALBEF as target, and why image captioning is only done with BLIP?
* Can the generated adversarial examples fool large models like llama 3.2, Claude, and GPT-4o?

**Details Of Ethics Concerns:**

The paper presents an advanced attack method against vision-language models.

---

> ### Author Response · Authors · 2024-11-25
> **Response to Reviewer y1vj (1/3)**
>
> We would like to thank the reviewer for the thoughtful comments on our paper. We  have carefully addressed the concerns and made revisions to improve the paper. We welcome any further suggestions or questions you may have and look forward to addressing them. Below are our response to the questions and weakness.
>
> **Q1**: Please explain why BLIP is excluded from I2T and T2I retrieval experiments, why visual grounding is only done with ALBEF as target, and why image captioning is only done with BLIP?
>
> **RQ1**: We evaluated the attack performance on I2T and T2I retrieval tasks across a variety of models, including CLIP with different backbones, ALBEF, and TCL, consistently demonstrating the effectiveness of the proposed method. To further illustrate its superiority, we have **included additional results on BLIP** for image-text retrieval in the appendix, highlighted in blue and annotated as R2Q1.
> The experiments on visual grounding were limited to ALBEF, and those on image captioning to BLIP, due to the unavailability of released checkpoints for other models. To ensure reproducibility and reliability, we only use publicly available VLP models. Furthermore, to demonstrate the transferability of the proposed method, we have added results on large vision-language models, including MiniGPT-4 and Llama 3.2, in image captioning and visual question answering tasks. **These results are detailed in Section A2.1 of the revised paper and highlighted in blue.**
>
> **Q2:** Can the generated adversarial examples fool large models like llama 3.2, Claude, and GPT-4o?
>
> **RQ2**: The proposed method is designed to promote adversarial transferability of attacks that can mislead various models that do not appear during training. To test whether the attack methods can fool black-box large vision-language models, we selected two open-source large vision-language models: MiniGPT-4 [1] and Llama 3.2 [2] to ensure reproducibility of our experiments. Specifically, MiniGPT-4, based on Vicuna V0 13B (a 13-billion-parameter large language model), has recently scaled up the capabilities of large language models and demonstrates performance comparable to GPT-4. For Llama 3.2, we utilized Llama-3.2-11B-Vision-Instruct, which comprises 11 billion parameters.
> In the image captioning task, we provided images with the prompt: “Describe this image, bringing all the details” as input. For the VQA task, we paired images with the question: “What is the content of this image?” and collected the generated descriptions and answers for each attack method.
>
> To assess attack performance, we used the CLIP score [3], which measures the similarity between the features of the generated descriptions/answers for adversarial images and their clean counterparts, as processed by the CLIP text encoder. To ensure fair comparisons, we calculated the CLIP score between the adversarial image features (generated by each attack method) and the clean image features, as large vision-language models may produce different responses to the same input at different query times. Additionally, we reported a baseline CLIP score by comparing the features of clean images with those of their randomly shuffled counterparts, providing a reference point to highlight the effectiveness of the attacks.

---

> ### Author Response · Authors · 2024-11-25
> **Response to Reviewer y1vj (2/3)**
>
> The results, provided below, demonstrate that our proposed method achieves the best performance across both tasks, further validating its transferability. However, we note that attack performance is relatively modest due to the substantial differences between the source and target models in terms of architecture, training data, and methodologies. Improving attack performance on large-scale models remains an open challenge, which we identify as a promising direction for future research.
> **Detailed experimental information, including settings, results, and corresponding analysis, has been added to Section A.3 of the revised paper. These additions are highlighted in blue and labelled as R5Q2.**
>
> Table 1. Evaluating robustness of Llama 3.2 in vision question answering, different CLIP text encoders are used to generate text features for similarity measurement
>
> | Method    | CLIP_ViT-B/16 | CLIP_ViT-B/32 | CLIP_ ViT-L/14 | CLIP_ RN50 | CLIP_ RN101| Average|
> |:---------:|:-----:|:------:|:-----:|:------:|:-----:|:------:|
> | Random Shuffle | 0.415| 0.417 | 0.312 | 0.380|   0.507| 0.406|
> | Co-Attack| 0.817 | 0.814  | 0.784| 0.802|   0.840 | 0.811  |
> | SGA | 0.815 | 0.811 | 0.783|   0.801| 0.839|   0.810|
> | ETU| 0.823  | 0.820| 0.792|   0.810| 0.845| 0.818|
> | VLATTACK| 0.812 | 0.809| 0.781| 0.799|   0.838| 0.808 |
> | VLPTransferAttack | 0.810  | 0.806| 0. 776| 0. 794|   0. 835| 0. 804|
> | MAA| 0. 798| 0. 794| 0. 763| 0. 782|   0. 825| 0. 793|
>
> Table 2. Evaluating robustness of MiniGPT-4 in image captioning, different CLIP text encoders are used to generate text features for similarity measurement
>
> | Method    | CLIP_ViT-B/16 | CLIP_ViT-B/32 | CLIP_ ViT-L/14 | CLIP_ RN50 | CLIP_ RN101| Average|
> |:---------:|:-----:|:------:|:-----:|:------:|:-----:|:------:|
> | Random Shuffle | 0. 445| 0. 427| 0. 302| 0. 398|   0. 631| 0. 441|
> | Co-Attack | 0. 856| 0. 823| 0. 771| 0. 847|   0. 875| 0. 834|
> | SGA | 0. 862| 0. 818| 0. 775|   0. 842| 0. 869|   0. 834|
> | ETU| 0. 880| 0. 825| 0. 792|   0. 862| 0. 883| 0. 848|
> | VLATTACK| 0. 853| 0. 815| 0. 780| 0. 843|   0. 874| 0. 833|
> | VLPTransferAttack | 0. 852| 0. 811| 0. 776| 0. 842|   0. 870| 0. 830|
> | MAA| 0. 843| 0. 801| 0. 768| 0. 833|   0. 859| 0. 821|
>
> [1] Deyao Zhu, Jun Chen, Xiaoqing Shen, Xiang Li, and Mohamed Elhoseiny. Minigpt-4: Enhancing vision-language understanding with advanced large language models. arXiv preprint arXiv:2304.10592, 2023.
>
> [2] Touvron, Hugo, et al. "Llama: Open and efficient foundation language models." arXiv preprint arXiv:2302.13971 (2023).
>
> [3] Zhao, Yunqing, Tianyu Pang, Chao Du, Xiao Yang, Chongxuan Li, Ngai-Man Man Cheung, and Min Lin. "On evaluating adversarial robustness of large vision-language models." Advances in Neural Information Processing Systems 36 (2024).

---

> ### Author Response · Authors · 2024-11-25
> **Response to Reviewer y1vj (3/3)**
>
> **W1**: The proposed method requires more computation power (iterations), which would either increase the delay of the attack (making it less ideal for real-time attacks), or require more powerful computers
>
> **RW1**: The proposed method addresses highly complex black-box application scenarios where target models and tasks are unknown. In various tasks, models focus on different types of information contained in the data. For instance, in image-retrieval tasks, models aim to match entire images to text, while in visual grounding and image captioning tasks, they emphasize fine-grained details such as objects or locations within the visual input that correspond to textual descriptions. This necessitates that attackers consider both global information and fine-grained details.
>
> To address this, we propose leveraging RScrop, which performs resizing and sliding cropping of images, alongside the multi-granularity similarity disruption (MGSD) strategy. These methods disrupt intra- and inter-model relationships across different granularities and hierarchical levels. For example, through the sliding operation of RScrop, the proposed approach effectively considers local regions and their dependencies to generate adversarial examples (evidence of this is provided in Section A.2). This significantly enhances the ability to prevent target models from recognizing visual elements.
>
> While this process requires more iterations and time compared to existing methods, the trade-off is acceptable. For example, on a workstation with a Tesla V100 GPU, the proposed method takes approximately 13.8 seconds to generate an adversarial image, whereas the SGA method requires 2.5 seconds. However, the proposed method achieves substantial performance gains—for instance, an improvement of approximately 40.6% in image-text retrieval—making the additional computational cost justifiable. In addition, SGA utilizes three-stage learning for adversarial text generations, incurring additional computational costs.
> Real-time attacks remain challenging for all current attack methods, as they typically optimize multiple adversarial examples in a batch and require several iterations [4–6]., which increases computational costs and reduces efficiency for real-time applications We plan to explore real-time optimization techniques in future work to generate adversarial examples for individual data points more efficiently.
>
> [4] Lin, Jiadong, et al. "Nesterov Accelerated Gradient and Scale Invariance for Adversarial Attacks." ICLR, 2019.
>
> [5] Chen, Huanran, et al. "Rethinking Model Ensemble in Transfer-based Adversarial Attacks." ICLR, 2023.
>
> [6] Huang, Zhichao, and Tong Zhang. " Black-box adversarial attack with transferable model-based embedding." ICLR, 2020.
>
> **W2**: Performance on BLIP seems not as good
>
> **RW2**: The proposed method focuses on challenging black-box attacks where target models and tasks are unknown. This scenario is particularly difficult because different models emphasize varying aspects of the data, and these disparities negatively impact the transferability of attacks. For example, an image-retrieval model focuses primarily on matching entire images to text, while an image captioning model aims to generate descriptive text that reflects specific objects and contextual information in an image. Successfully misleading such models requires attackers to identify and manipulate objects, attributes, and contextual details effectively.
> Even in this complex scenario, the proposed method outperforms compared methods. For example, we use CLIP_ViT-B/16 for image-text retrieval as the source model to generate adversarial examples for testing the robustness of BLIP in image captioning. The results demonstrate that compared methods perform worse, and the proposed method achieves approximately an 8.37% improvement over them. Furthermore, **as shown in Table 8 and Figure 4 of the revised paper**, the proposed method captures more fine-grained details, which enhances attack performance.
>
> **W3**: The writing could be improved. For example, I don't think formula (1) matches its description.
>
> **RW3**: Thank you for the valuable suggestion. The intention behind formula (1) is to demonstrate how the proposed method increases the distance between clean and adversarial images across different layers. For vision transformer-based VLP models, features include outputs from each self-attention module and the final output layer, while for ResNet-based VLP models, features are extracted from each residual block and the final output layer.
>
> In the revised version, we have updated formula (1) to more accurately reflect the underlying concept and provided a clearer description of the algorithm to ensure better alignment between the formula and its explanation. **This change has been highlighted in blue and labelled as R2W3**.

---

> ### Author Response · Authors · 2024-11-28
> **A Kind Reminder: Follow-Up on Revisions and Responses**
>
> Dear Reviewer y1vj,
>
> Thank you once again for your time and insightful suggestions on our paper!
>
> As the discussion period comes to an end, we would like to check whether our responses and the revised paper have adequately addressed your concerns. Your feedback is invaluable to us, and we would greatly appreciate the opportunity to engage in further discussions if needed.
>
> Sincerely, The Authors

---

> > ### Comment · Reviewer_y1vj · 2024-12-02
> >
> > Dear authors,
> >
> > Thank you for the clarifications and new experiment results. I'm satisfied with most of the responses, except that I don't find MAA performs the best on Llama-3.2 and MiniGPT-4. I hope to see a short discussion on why MAA does not perform as well on larger models like Llama-3.2, "substantial differences between the source and target models in terms of architecture, training data, and methodologies" is too vague and is not a good explanation. Something like what is in the abstract: "mainly attributed to the over-reliance on model-specific features and regions, particularly in the image modality" is more concrete.

---

> ### Author Response · Authors · 2024-12-02
> **Response to Official Comment by Reviewer y1vj (1/2)**
>
> Thank you for your response! Your feedback is invaluable in improving our paper. Our clarifications for the questions are outlined below:
>
> RE 1. Performance on Large Vision-Language Models.
>
> In experiments on large vision-language models, we use the CLIP score to measure the similarity between the features of the generated descriptions/answers for adversarial images and their clean counterparts. A **lower CLIP score** indicates a **larger distance** between the generated descriptions/answers for adversarial images and their clean counterparts, meaning an attack model is **better** at altering the semantics of the original images. The results show that the proposed method achieves the **lowest score** across all compared methods, indicating it achieves the **best performance**. The **Random Shuffle** method compares the features of generated texts of clean images with those of their randomly shuffled counterparts. It serves as **a reference point** for all methods.
>
> RE 2. Reasons of limited transferability to Large Vision-Language Models.
>
> The low transferability between **general vision-language models** and **large vision-language models**, e.g., CLIP to LLaMA 3.2, is due to the reliance on model-specific features and the varying generalization abilities, which arise from differences in model parameters, architectures, training data, and loss functions. Large vision-language models, such as LLaMA 3.2, have more parameters, deeper architectures, and are trained on much larger datasets compared to general vision-language models. For instance, **LLaMA 3.2**, which we use in our experiments, has **billions of parameters **and is trained on **billions of image-text pairs**, whereas CLIP models have **millions of parameters** and are trained on **millions of data pairs**. Furthermore, they employ different training losses: LLaMA is trained with cross-entropy loss, while CLIP is primarily trained using contrastive loss. On the one hand, these differences make the models reliant on some distinct, specific features to make decisions. On the other hand, models with more parameters, deeper architectures, and more training data typically exhibit **better generalization abilities** and are more robust to adversarial perturbations, making attacks less likely to transfer successfully from general vision-language models to large language models. The low transferability observed in our experiments supports this. **We will include these analysis in the revised version of our paper.**

---

### Official Review · Reviewer_pquT · 2024-11-11

**Soundness:** 3
**Presentation:** 3
**Contribution:** 3
**Rating:** 5
**Confidence:** 4

**Summary:**

This paper introduces the Meticulous Adversarial Attack (MAA) to address the limited transferability of current adversarial attacks on vision-language pretrained (VLP) models. MAA combines the Resizing and Sliding Crop (RScrop) technique to diversify adversarial examples and the Multi-Granularity Similarity Disruption (MGSD) strategy to enhance embedding distances across different model layers. Extensive experiments demonstrate that MAA improves both the effectiveness and transferability of adversarial attacks across various VLP models, datasets, and tasks.

**Strengths:**

1.	MAA enhances spatial coherence and model sensitivity by focusing on fine-grained local details through scaling and sliding techniques. It improves adversarial transferability by generating model-generic examples based on shared low-level features.
2.	MAA outperforms existing methods in adversarial transferability and provides valuable insights for adversarial attacks and defense on VLP through extensive parameters, studies and discussion.

**Weaknesses:**

1.	The author lacks citations and evidence for some of the viewpoints in the paper. In line 78, why are VLP models more prone to relying on specific features and regions to associate images and texts? And in line 101, why do vision transformers tend to lose the crucial contextual information in the image? As far as I’m concerned, the vision transformer ensures that each patch embedding contains features from neighboring regions through its self-attention mechanism.
2.	The author needs to add experiments and visualization results to demonstrate that the application of the RScrop technique helps extract finer-grained contextual features from the image.
3.	I suggest that the author explore ensembling multiple source models to further improve transferability and attempt to apply adversarial attacks on larger black-box models, such as GPT-4.

**Questions:**

As mentioned in Weaknesses.

---

> ### Author Response · Authors · 2024-11-25
> **Response to Reviewer pquT (1/4)**
>
> We would like to thank the reviewer for the thoughtful feedback and valuable suggestions. We have carefully addressed the comments and made revisions to improve the paper. Below, we provide our responses to the reviewer’s questions and the corresponding changes in the manuscript. We appreciate the opportunity to clarify and improve our work.
>
> **W1.1**: why are VLP models more prone to relying on specific features and regions to associate images and texts.
>
> **RW1.1**: During the learning process, the sub-networks of a multi-modal model for each modality tend to focus on modality-specific features for their predictions. This tendency increases the likelihood that the model relies heavily on certain features for image-text associations [1,2].
>
> From the perspective of perturbations, this phenomenon becomes even more pronounced. Perturbations generated by standard methods, which are effective in misleading target models in white-box settings, often fail to transfer to other models. This is reflected in our experiments (see Table 2 and 3), where overfitting to the source model results in significant performance degradation. The low transferability observed suggests that different models process data from distinct perspectives, further supporting the idea that VLP models are prone to relying on specific features.
>
> To address this issue, we propose exploring more fine-grained details to better understand image content. This approach prevents overfitting to model-specific features and promotes transferable attacks. Our experiments show that the proposed method achieves improved performance across different models and tasks.
>
> **W1.2**: In line 101, why do vision transformers tend to lose the crucial contextual information in the image?
>
> **RW1.2**: Vision Transformers (ViTs) do not inherently possess inductive biases such as spatial locality and translation invariance, as highlighted in recent studies [5-8]. Standard ViTs process images by dividing them into fixed-size patches, treating each patch as a token. This token-based approach limits the model's ability to capture object features at varying scales and locations, as well as the essential local structures formed by neighboring pixels [3-7]. As a result, important contextual information within and across patches is not fully leveraged. Our experimental results show that compared methods exhibit lower transferability across different models, suggesting that they fail to effectively capture the characteristics and vulnerabilities of the samples, leading to missed important information.
>
> To further demonstrate this, we conduct **additional experiments and provide visualizations**, which are presented in **Section A.2**. These results show that standard vision transformers lose crucial contextual information, while the proposed method helps to capture this information more effectively. Further experimental results can also be found in our response to the second weakness you raised.
>
> [1] Wang, Weiyao, Du Tran, and Matt Feiszli. "What makes training multi-modal classification networks hard?." Proceedings of the IEEE/CVF conference on computer vision and pattern recognition. 2020.
>
> [2] Yao, Y., & Mihalcea, R. (2022, May). Modality-specific learning rates for effective multimodal additive late-fusion. In Findings of the Association for Computational Linguistics: ACL 2022 (pp. 1824-1834).
>
> [3] Maurício, José, Inês Domingues, and Jorge Bernardino. "Comparing vision transformers and convolutional neural networks for image classification: A literature review." Applied Sciences 13.9 (2023): 5521.
>
> [4] Han, Kai, et al. "Transformer in transformer." Advances in neural information processing systems 34 (2021): 15908-15919.
>
> [5] Yuan, Li, et al. "Tokens-to-token vit: Training vision transformers from scratch on imagenet." Proceedings of the IEEE/CVF international conference on computer vision. 2021.
>
> [6] Wu, Haiping, et al. "Cvt: Introducing convolutions to vision transformers." Proceedings of the IEEE/CVF international conference on computer vision. (2021): 22-31.
>
> [7] Hatamizadeh, Ali, Hongxu Yin, Greg Heinrich, Jan Kautz, and Pavlo Molchanov. "Global context vision transformers." In ICML, 2023: 12633-12646.

---

> ### Author Response · Authors · 2024-11-25
> **Response to Reviewer pquT (2/4)**
>
> **W2**: The author needs to add experiments and visualization results to demonstrate that the application of the RScrop technique helps extract finer-grained contextual features from the image.
>
> **RW2**: Thank you for the suggestion. The proposed method leverages the RScrop technique to extract finer-grained details, enabling a better understanding of the characteristics and vulnerabilities of samples. Through extensive experiments, we demonstrate that our method achieves superior performance compared to baseline methods, indicating its ability to capture more detailed contextual information.
>
> To further substantiate this, we conducted experiments on ALBEF model in visual grounding, **comparing its performance with and without the proposed augmentation**. The evaluation metric measures the overlap between the predicted and ground-truth regions. If the overlap exceeds a certain threshold (e.g., 0.5), the regions are considered matched. The results are summarized below:
>
> Table 1.Performance on visual grounding.
> | | Val | TestA | TestB |
> |:----:|:-----:|:------:|:-----:|
> | Model w/o the proposed method | 51.24 | 56.71   | 44.79 |
> | Model w/ the proposed method |  56.67 | 65.37| 45.28 |
>
> These results clearly show that the proposed method improves the overlap with ground-truth regions, demonstrating its ability to extract finer-grained contextual features.
>
> To provide additional evidence, we applied **Grad-CAM visualizations to highlight the activation maps of images**, indicating the contribution of different regions to the model’s predictions. Examples of these visualizations are presented in **Figure 4** of the revised paper. From the visualizations, it can be observed that the proposed method captures finer-grained contextual features more effectively than the baseline model. Specifically, key observations can be seen as follows:
>
> (1). Comprehensive region coverage: The proposed method focuses on image regions relevant to the caption. For example, it captures the clothes in image (a), the area contributing to the bending behavior in image (c), and the entirety of the motorcycle in image (d).
>
> (2). Contextual awareness: The method identifies cohesive areas, such as the faces and clothing of the man in image (b), the cat in image (e), and the pitcher in image (g). This demonstrates the improved contextual awareness of our approach.
>
> (3). Enhanced relational understanding: For instance, in image (d), the proposed method captures features for both motorcycles, reflecting an improved understanding of relationships between objects described in the caption.
>
> (4). Broad object focus: The proposed method emphasizes the entire area of relevant objects rather than isolating specific regions. For example, in image (a), it highlights multiple areas of the bed instead of concentrating on just a few regions.
>
> These results highlight that the proposed method effectively extracts fine-grained details and contextual information, thereby enabling a deeper exploration of the characteristics and vulnerabilities of samples, which ultimately enhances attack performance.
>
> We have incorporated the experimental results into **Table 8**, added the visualizations into **Figure 4**, and provided detailed analysis in the revised paper. **These updates are highlighted in blue and labelled as R1W2.**

---

> ### Author Response · Authors · 2024-11-25
> **Response to Reviewer pquT (3/4)**
>
> **W3**: I suggest that the author explore ensembling multiple source models to further improve transferability and attempt to apply adversarial attacks on larger black-box models, such as GPT-4.
>
> **RW3**: Thank you for the suggestion.
>
> **(1) For ensemble attacks**, we explored **various combinations of source models**, including: ensembles of **homogeneous** CLIP models, e.g., CLIP_RN50 + CLIP_RN101, ensembles of **heterogeneous** models, e.g., CLIP_ViT-B/16 + CLIP_RN101, and ensembles of models of **different architectures and training process**, e.g., CLIP + ALBEF. The results are shown in the following table and have also been included in Section A.1 of the revised paper.
>
> Table 1. The attack success rate (%, R@1) of utilizing different ensemble source models in image-text retrieval on Flickr30K.
>
> | Target Model | CLIP_ViT-B/16 | CLIP_ ViT-L/14 | CLIP_ RN50| CLIP_ RN101 | ALBEF | TCL|
> |:-------------:|:-----:|:------:|:-----:|:------:|:------:|:------:|
> | Source Model |I2T, T2I| I2T, T2I. | I2T , T2I | I2T, T2I | I2T, T2I| I2T, T2I |
> | CLIP_ViT-B/16| 99.98,99.99| 54.36,63.02|   77.14,80.58|   73.05,74.85|   31.80,43.62| 41.10,49.81|
> | CLIP_ RN101|  36.81, 47.97| 30.80, 45.26|   70.50, 78.77|   98.34, 97.63| 13.56, 29.3  |   17.18, 31.62|
> | ALBEF| 39.14, 49.77| 32.39, 47.29  |   51.09 , 60.99 |  43.68, 54.99  | 100.00 , 99.95|   75.87,76.17|
> | TCL | 41.96, 50.71  | 34.60, 50.00|   55.30, 64.05  |    49.30,57.67|    72.78,75.65  | 100.00, 100.00|
> | CLIP_ViT-B/16 + CLIP_ ViT-L/14. | 99.63, 99.87   | 99.63, 99.87  |  62.45, 68.13  |    27.22 , 41.3 | 40.35 ,50.72  |   49.32 , 56.52  |
> | CLIP_ RN50 + CLIP_ RN101|  38.9    ,50.35  | 33.13,46.91  |   99.87,99.27  |   99.62, 99.83  | 15.24, 29.88|   18.34, 34.52|
> | ALBEF + TCL | 43.68, 53.23  | 41.89 , 53.14|   57.82 ,66.99|     24.81, 31.62    |    96.13, 99.27|     98.42, 99.43|
> | CLIP_ViT-B/16 + CLIP_ RN101| 99.02,98.32| 52.39, 62.37|   75.22, 80.27|  98.34 ,97.84|   30.87, 44.43|  36.14,49.02|
> | CLIP_ViT-B/16 + ALBEF | 99.75, 99.68| 60.74, 69.75|   76.63,79.31|    73.56, 75.09|    99.58, 99.56|    85.04, 84.88|
> | CLIP_ViT-B/16 + CLIP_ViT-L/14+ ALBEF | 97.67, 98.13| 98.65, 97.71|   77.39, 78.77|   73.95, 77.63|   96.77, 96.8|   81.66, 81.40|
>
> We provide an analysis of these results to better understand the effectiveness of the ensemble method. Specifically, we highlight the following observations: “On the one hand, using ensembles often improves performance, particularly when combining complementary models. For example, the combination of CLIP_ViT-B/16 + CLIP_ViT-L/14+ ALBEF and LIP_ViT-B/16 + ALBEF outperforms using a single source model when attacking ALBEF and TCL. This supports our assertion in the Related Work section that methods capable of attacking a set of models are more likely to mislead more models. One the other hand, utilizing more models does not always lead to better performance. For instance, CLIP_ViT-B/16 + CLIP_ RN101 performs worse than using CLIP_ViT-B/16 alone. This may be due to varying contributions of different models to the attack. In this case, CLIP_ViT-B/16 is generally more effective than CLIP_ RN101, and the less effective model may dilute the overall performance when combined.”
> We also discuss the limitations of this approach: “However, there are two main limitations to such a method. First, in many scenarios, diverse models might not be readily available. Second, utilizing multiple models significantly increases computational overhead, which can hinder practicability and scalability” .
>
> **All relevant content has been added to Section A.1  of the revised paper, highlighted in blue and labelled as R1W3.**

---

> ### Author Response · Authors · 2024-11-25
> **Response to Reviewer pquT (4/4)**
>
> **(2) Regarding attacks on larger models**, to ensure reproducibility, we selected two open-source large vision-language models: **MiniGPT-4** [8] and **Llama 3.2** [9].
>
> Specifically, MiniGPT-4, based on Vicuna V0 13B (a 13-billion-parameter large language model), has recently scaled up the capabilities of large language models and demonstrates performance comparable to GPT-4. For Llama 3.2, we utilized Llama-3.2-11B-Vision-Instruct, which comprises 11 billion parameters.
> In the image captioning task, we provided images with the prompt: “Describe this image, bringing all the details” as input. For the VQA task, we paired images with the question: “What is the content of this image?” and collected the generated descriptions and answers for each attack method.
>
> To assess attack performance, we used the CLIP score [10], which measures the similarity between the features of the generated descriptions/answers for adversarial images and their clean counterparts, as processed by the CLIP text encoder. To ensure fair comparisons, we calculated the CLIP score between the adversarial image features (generated by each attack method) and the clean image features, as large vision-language models may produce different responses to the same input at different query times. Additionally, we reported a baseline CLIP score by comparing the features of clean images with those of their randomly shuffled counterparts, providing a reference point to highlight the effectiveness of the attacks.
>
> The results, provided below, demonstrate that our proposed method achieves the best performance across both tasks, further validating its transferability. However, we note that attack performance is relatively modest due to the substantial differences between the source and target models in terms of architecture, training data, and methodologies. Improving attack performance on large-scale models remains an open challenge, which we identify as a promising direction for future research.
> **Detailed experimental information, including settings, results, and corresponding analysis, has been added to Section A.3 of the revised paper. These additions are highlighted in blue and labelled as R5Q2.**
>
> Table 3. Evaluating robustness of Llama 3.2 in vision question answering, different CLIP text encoders are used to generate text features for similarity measurement
>
> | Method    | CLIP_ViT-B/16 | CLIP_ViT-B/32 | CLIP_ ViT-L/14 | CLIP_ RN50 | CLIP_ RN101| Average|
> |:---------:|:-----:|:------:|:-----:|:------:|:-----:|:------:|
> | Random Shuffle | 0.415| 0.417 | 0.312 | 0.380|   0.507| 0.406|
> | Co-Attack| 0.817 | 0.814  | 0.784| 0.802|   0.840 | 0.811  |
> | SGA | 0.815 | 0.811 | 0.783|   0.801| 0.839|   0.810|
> | ETU| 0.823  | 0.820| 0.792|   0.810| 0.845| 0.818|
> | VLATTACK| 0.812 | 0.809| 0.781| 0.799|   0.838| 0.808 |
> | VLPTransferAttack | 0.810  | 0.806| 0. 776| 0. 794|   0. 835| 0. 804|
> | MAA| 0. 798| 0. 794| 0. 763| 0. 782|   0. 825| 0. 793|
>
> Table 4. Evaluating robustness of MiniGPT-4 in image captioning, different CLIP text encoders are used to generate text features for similarity measurement
>
> | Method    | CLIP_ViT-B/16 | CLIP_ViT-B/32 | CLIP_ ViT-L/14 | CLIP_ RN50 | CLIP_ RN101.| Average|
> |:---------:|:-----:|:------:|:-----:|:------:|:-----:|:------:|
> | Random Shuffle | 0. 445| 0. 427| 0. 302| 0. 398|   0. 631| 0. 441|
> | Co-Attack | 0. 856| 0. 823| 0. 771| 0. 847|   0. 875| 0. 834|
> | SGA | 0. 862| 0. 818| 0. 775|   0. 842| 0. 869|   0. 834|
> | ETU| 0. 880| 0. 825| 0. 792|   0. 862| 0. 883| 0. 848|
> | VLATTACK| 0. 853| 0. 815| 0. 780| 0. 843|   0. 874| 0. 833|
> | VLPTransferAttack | 0. 852| 0. 811| 0. 776| 0. 842|   0. 870| 0. 830|
> | MAA| 0. 843| 0. 801| 0. 768| 0. 833|   0. 859| 0. 821|
>
> [8] Deyao Zhu, Jun Chen, Xiaoqing Shen, Xiang Li, and Mohamed Elhoseiny. Minigpt-4: Enhancing vision-language understanding with advanced large language models. arXiv preprint arXiv:2304.10592, 2023.
>
> [9] Touvron, Hugo, et al. "Llama: Open and efficient foundation language models." arXiv preprint arXiv:2302.13971 (2023).
>
> [10] Zhao, Yunqing, Tianyu Pang, Chao Du, Xiao Yang, Chongxuan Li, Ngai-Man Man Cheung, and Min Lin. "On evaluating adversarial robustness of large vision-language models." Advances in Neural Information Processing Systems 36 (2024).

---

> ### Author Response · Authors · 2024-11-28
> **A Kind Reminder: Follow-Up on Revisions and Responses**
>
> Dear Reviewer pquT,
>
> Thank you once again for your time and insightful suggestions on our paper!
>
> As the discussion period comes to an end, we would like to check whether our responses and the revised paper have adequately addressed your concerns. Your feedback is invaluable to us, and we would greatly appreciate the opportunity to engage in further discussions if needed.
>
> Sincerely,
> The Authors

---

### Author Response · Authors · 2024-11-25
**Response Summary**

We would like to express our sincere gratitude to the reviewers for their insightful and constructive feedback. We deeply appreciate their recognition of the strengths of our work, including the originality and significance of the proposed method (Reviewers #pquT, #PuP3, #cGtF), the valuable insights it provides to the research field (Reviewers #pquT, #PuP3), the thorough experimental setup and comprehensive evaluation (Reviewers #y1vj, #3fPG, #cGtF), and the clear presentation (Reviewers #3fPG, #cGtF).
The reviewers' suggestions have offered valuable guidance for enhancing the quality and scope of our paper, and we have made substantial efforts to address these points by refining the manuscript, adding in-depth analysis, and conducting comprehensive experiments. Specifically, we have:

1. Conducted additional experiments to explore the use of ensembling multiple source models (Reviewer #pquT) and compared the proposed method with diverse data augmentation techniques to demonstrate the effectiveness of the proposed method (Reviewer #PuP3).

2. Evaluated the proposed method on representative large vision-language models, such as Llama 3.2 and MiniGPT-4 (Reviewers #pquT, #y1vj), and in additional tasks (Reviewer #cGtF) to demonstrate its applicability.

3. Included comparisons with more recent works to provide a broader context (Reviewer #cGtF).

4. Added citations and evidence to support key viewpoints in the paper (Reviewer #pquT) and provided in-depth analysis of the proposed methods (Reviewer #PuP3).

5. Incorporated results using different random seeds to ensure the validity and stability of the experimental outcomes (Reviewer #3fPG).

6. Improved the presentation to eliminate typos and enhanced the clarity and understanding of both the research field and the proposed method (Reviewers #PuP3, #cGtF, #3fPG), and discussed the limitations of the proposed method and potential future research directions (Reviewer #cGtF).

These thoughtful suggestions have significantly contributed to refining the clarity, depth, and impact of our work, and we are committed to fully incorporating them into the revised version of the paper. The revised paper will be uploaded shortly, as there are many contents that need to be added and updated.

---

### Author Response · Authors · 2024-11-28
**Happy to take  follow-up questions before the author-reviewer discussion cloes**

Dear Reviewers and ACs,

Thank all for your time and insightful suggestions on our paper!

As the discussion period comes to an end, we would like to check whether our responses and the revised paper have adequately addressed your concerns. Your feedback is invaluable to us, and we would greatly appreciate the opportunity to engage in further discussions if needed.

Sincerely,
The Authors

---

### Meta-Review · Area_Chair_tZZH · 2024-12-20

**Metareview:**

This paper introduces Meticulous Adversarial Attack (MAA), a method designed to enhance the transferability and effectiveness of adversarial attacks on vision-language pre-trained (VLP) models by leveraging model-independent vulnerabilities. Through techniques like resizing and sliding crop (RScrop) and multi-granularity similarity disruption (MGSD), MAA generates diverse adversarial examples and maximizes embedding distance, demonstrating improvements in attack generalizability across diverse models, datasets, and tasks. During the rebuttal phase, the authors provided additional experiments to address reviewers' concerns. The Area Chair and the reviewers engaged in further discussions regarding the contribution of this work and agreed that the novelty of the proposed method is limited, as it primarily relies on data augmentation (RScrop) and multiple-layer outputs. In the additional experiments on larger vision-language models like Llama 3.2 and MiniGPT-4, the improvement in attack success rates was marginal. Therefore, we have decided not to accept the work in its current state.

**Additional Comments On Reviewer Discussion:**

We carefully reviewed the authors' additional experiments and explanations addressing the reviewers' concerns and agree that the novelty of this work and its contribution to the field are limited.

---

### Decision · Program_Chairs · 2025-01-22

Reject